# New insights into the cellular temporal response to proteostatic stress

Justin Rendleman[1†], Zhe Cheng[1†], Shuvadeep Maity[1†], Nicolai Kastelic[2], Mathias Munschauer[2], Kristina Allgoewer[1], Guoshou Teo[1], Yun Bin Matteo Zhang[1], Amy Lei[1], Brian Parker[1], Markus Landthaler[2,3], Lindsay Freeberg[4], Scott Kuersten[4], Hyungwon Choi[5,6], Christine Vogel[1*]

[1]Center for Genomics and Systems Biology, Department of Biology, New York University, New York, United States; [2]Berlin Institute for Medical Systems Biology, Max Delbrück Center for Molecular Medicine, Berlin, Germany; [3]Integrative Research Institute for the Life Sciences, Institute of Biology, Humboldt University, Berlin, Germany; [4]Illumina Inc, Madison, United States; [5]National University of Singapore, Singapore; [6]Institute of Molecular and Cell Biology, Agency for Science, Technology and Research, Singapore

**Abstract** Maintaining a healthy proteome involves all layers of gene expression regulation. By quantifying temporal changes of the transcriptome, translatome, proteome, and RNA-protein interactome in cervical cancer cells, we systematically characterize the molecular landscape in response to proteostatic challenges. We identify shared and specific responses to misfolded proteins and to oxidative stress, two conditions that are tightly linked. We reveal new aspects of the unfolded protein response, including many genes that escape global translation shutdown. A subset of these genes supports rerouting of energy production in the mitochondria. We also find that many genes change at multiple levels, in either the same or opposing directions, and at different time points. We highlight a variety of putative regulatory pathways, including the stress-dependent alternative splicing of aminoacyl-tRNA synthetases, and protein-RNA binding within the 3' untranslated region of molecular chaperones. These results illustrate the potential of this information-rich resource.
DOI: https://doi.org/10.7554/eLife.39054.001

*For correspondence: cvogel@nyu.edu

†These authors contributed equally to this work

## Introduction

Proteostasis, the integrated assembly of pathways that regulate transcription and translation, the proteins' correct folding, localization, and - eventually - degradation, is a major challenge for the cell (*Balch et al., 2008*; *Labbadia and Morimoto, 2015*). Much of the cellular energy expenditure is dedicated to maintaining a healthy proteome (*Buttgereit and Brand, 1995*), as millions of molecules are produced every single minute (*Harper and Bennett, 2016*). Challenges to proteostasis, that is the accumulation of unfolded or misfolded proteins in the endoplasmic reticulum (ER), are key to human diseases, including cancer, diabetes, and neurodegeneration (*Zhao and Ackerman, 2006*; *Morimoto, 2013*; *Labbadia and Morimoto, 2015*).

Therefore, the Unfolded Protein Response (UPR), triggered by challenged proteostasis, affects all levels of gene expression regulation creating an intricate and highly dynamic network of responses (*Figure 1A*). Initially the ER stress sensor *PERK* is autophosphorylated leading to the subsequent phosphorylation of the initiation factor eIF2α limiting its use for translation initiation, thereby decreasing the ER's protein folding load (*Wek and Cavener, 2007*; *Harding et al., 1999*; *Ron and Walter, 2007*).

**Figure 1.** Protein misfolding stress involves multiple processes. (**A**) We investigate the multi-layered regulation during the response to protein misfolding stress. The schematic illustrates the simplified relationship between protein misfolding, stress of the endoplasmic reticulum (ER), and oxidative stress. Tunicamycin elicits ER stress, which triggers various downstream effects including transcription, translation, and RNA and protein degradation. Attempts to refold proteins increases production of hydrogen peroxide in the cell. Hydrogen peroxide, in turn, elicits oxidative stress through an imbalance of reactive oxygen species (ROS). In cancer cells, basal ROS levels can be heightened due to altered metabolism. (**B**) Our experiment extracts significant regulatory events for >7,000 genes in response to either tunicamycin or hydrogen peroxide treatment. The experimental design maps multiple layers of regulation in response to stress, with an emphasis on post-transcriptional regulation. RNA and protein abundances were measured using RNA-seq and mass spectrometry, respectively. Ribosome footprinting and protein occupancy profiling were used to map the binding of ribosomes and non-ribosomal proteins along mRNAs, respectively. Time points and genes with significant regulation were extracted from each data type with the PECA tool (*Teo et al., 2018*; *Teo et al., 2014*). The heatmap and PECA results show example genes: the stress response genes *HSPA5* (GRP78, BiP) and *HSP90B1*, the DNA repair gene *RAD51*, and the aminoacyl-tRNA synthetase *WARS*. FDR - false discovery rate.

DOI: https://doi.org/10.7554/eLife.39054.002

The following figure supplement is available for figure 1:

**Figure supplement 1.** Overview of the statistical workflow for data analysis (PECA).

*Figure 1 continued on next page*

*Figure 1 continued*

DOI: https://doi.org/10.7554/eLife.39054.003

Global translation suppression is accompanied by the targeted activation of translation for important UPR regulators and the transcription of downstream stress-response genes. The best characterized example of such translation induction in mammalian systems is the transcription factor *ATF4*, where increased eIF2α phosphorylation allows ribosomes to bypass inhibitory upstream open reading frames (uORFs) in the 5' untranslated region (UTR) during stress (*Vattem and Wek, 2004*).

Another branch of the mammalian UPR includes the major transcription factor, *XBP1*, which is activated through non-canonical splicing by the *IRE1* endonuclease. *IRE1* also targets mRNAs for degradation, further relieving the ER from the burden of protein synthesis (*Calfon et al., 2002*; *Hollien et al., 2009*). A third UPR transcription factor, *ATF6*, is released from the ER membrane upon accumulation of misfolded proteins and activated through proteolytic cleavage in the Golgi (*Haze et al., 1999*). Finally, the proteasome degrades irreparably damaged and ubiquitinated proteins (*Plemper and Wolf, 1999*) - demonstrating how indeed, over the course of several hours, the cell systematically responds to ER stress at all levels of gene expression regulation to adapt to the new conditions.

This elaborate regulatory network encompasses many questions that still remain unanswered, illustrating the need for integrative assessment of the coordination amongst these different processes. One such question involves the relationship between these response pathways, both across genes as well as over time. A specific gene may be regulated by multiple pathways over the entire course of the stress response; additionally, these pathways might act concordantly, discordantly, simultaneously or sequentially, thus adding to the complexity of possible gene expression regulation. The dynamics of the stimulus determines the transition between transient and chronic stress, as well as the reactivation of translation and the cell's decision between survival and apoptosis (*Brush et al., 2003*; *Guan et al., 2017*; *Guan et al., 2014*; *Woehlbier and Hetz, 2011*; *Li et al., 2010*; *Quirós et al., 2017*). Establishing the factors responsible for this decision is essential to understanding how aberrant proteostasis leads to disease. Further, while a short list of uORF regulated genes is known to escape translation inhibition similar to *ATF4*, recent studies suggest more genes are likely to be positively regulated at the level of translation during ER stress (*Baird et al., 2014*; *Cullinan et al., 2003*; *Guan et al., 2014*; *Maity et al., 2016*; *Ventoso et al., 2012*). However, the extent and importance of this translation upregulation is largely unknown.

Translation regulation is not only linked to the ER and proteostatic stress, for example through eIF2α phosphorylation, but it is also intimately linked to energy metabolism (*Leibovitch and Topisirovic, 2018*; *Buttgereit and Brand, 1995*; *Wang et al., 2011*). Conversely, depletion in metabolic energy can trigger the formation of misfolded proteins (*Grootjans et al., 2016*; *Kaufman, 2002*; *Ron, 2002*; *Pavitt and Ron, 2012*). The resulting challenge is particularly prominent in cancer cells, as they are subject to both elevated energy needs and increased protein synthesis - yet, we are only beginning to understand how the cell balances these demands (*Leibovitch and Topisirovic, 2018*; *Hazari et al., 2016*).

Finally, ER stress and the UPR are interconnected with the oxidative stress response, largely due to reactive oxygen species (ROS) produced during protein folding in the ER (*Malhotra and Kaufman, 2007*)(*Figure 1A*). In addition, many cancer cells have inherently higher basal ROS levels due to the challenges mentioned above (*Liou and Storz, 2010*), which can lead to the activation of the UPR to promote tumor growth (*Yadav et al., 2014*). Both ER and oxidative stress have common elements in their elicited cellular response, however the extent of this shared stress response and is unclear.

To provide new insights into these open questions we conducted one of the most comprehensive assessments of the regulatory landscape in response to ER stress available to date - revealing previously unknown and underappreciated aspects of the adaptive UPR. We collected replicate samples at four time points (0, 1, 4, and 8 hr) from human cervical cancer cells treated with tunicamycin. Tunicamycin is a naturally occurring antibiotic that inhibits protein N-linked glycosylation and therefore prevents proper folding particularly in the ER, thus inducing the accumulation of misfolded proteins (*Mahoney and Duksin, 1979*; *Noda et al., 1999*). Using RNA-seq and mass spectrometry, we

determine the complete RNA and protein concentrations for >7,000 genes in the core dataset and for >14,000 genes in the extended data. Further, using ribosome footprinting and protein-occupancy profiling, we map the interactome of ribosomes and non-ribosomal proteins to mRNA, which informs on translation as well as several aspects of RNA processing, respectively. Finally, we apply the same technologies to cells treated with hydrogen peroxide to elicit oxidative stress and compare the responses. The presented data explores both shared and stress-specific regulation, it describes both the early and later stress response, and it disentangles transcriptional from post-transcriptional regulation. We discuss general patterns seen across the core set of genes and explore specific examples that illustrate new pathways important in the cell's adaptation to stress.

## Results

### Multi-layered data types identify new regulatory signatures during stress

We present a resource comprising four complementary data types collected for cervical cancer cells treated with tunicamycin and hydrogen peroxide: RNA and protein concentrations measured with RNA-seq and quantitative proteomics, respectively, and the binding profiles of ribosomes and non-ribosomal proteins along mRNAs measured with ribosome footprinting and protein occupancy profiling (*Figure 1A*). The data provides a comprehensive picture of the regulatory landscape in response to stress, with a specific focus on post-transcriptional regulation during the Unfolded Protein Response. Ribosome footprints mapping to coding regions serve as an estimate of translation efficiency (*Ingolia et al., 2011*; *Ingolia et al., 2009*). In comparison, we use protein occupancy profiling to monitor footprints of other, non-ribosomal proteins on the untranslated regions (UTRs) (*Baltz et al., 2012*; *Castello et al., 2016*). These proteins are candidate regulators that bind to the UTRs to alter translation, RNA stability, and localization. While protein occupancy profiling, similar to ribosome footprinting, provides high-resolution, single-nucleotide level data, it does not reveal the identity of the bound proteins.

To account for the dynamic nature of the stress response, we monitor four time points (0, 1, 4, 8 hr). As illustrated in *Figure 1A*, protein misfolding stress in cancer cells is tightly intertwined with the response to oxidative stress. Therefore, we also subjected the cells to oxidative stress enabling the identification of components of the shared stress response and those specific to the UPR. From the extended sequencing data comprising >14,000 genes with triplicate measurements and >10,000 quantified proteins (*Supplementary file 1–4*), we compile a core set of 7,011 genes with complete, replicate measurements (*Figure 3—figure supplement 5*). We identify general and specific regulatory signatures in this core set that provide many new insights into the adaptive response to stress.

As expression changes are interdependent, for example protein expression depends on RNA changes, we apply our tool for Protein Expression Control Analysis (PECA)(*Teo et al., 2018*; *Teo et al., 2014*) to extract significant regulatory events at specific levels (*Figure 1B*). PECA is designed to 'subtract' the change in transcript abundance for each gene to isolate the contribution of translation and degradation to protein expression changes (*Figure 1—figure supplement 1*). PECA also takes into account the temporal information to report both significantly regulated genes *and* time points (change points). We expand the use of PECA to all four datasets and convert concentration and binding data into information on regulatory events. For the RNA data, assuming DNA copy number does not change, PECA informs on significant changes in transcription and RNA degradation (TRXP/RNA-DEG)(*Figure 1B*). For ribosome footprinting data, PECA extracts information on translation (TRL) by controlling for changes at the RNA level. For the protein occupancy data, PECA extracts information on the binding of post-transcriptional regulators that can affect translation, RNA localization, and degradation (TRL/RNA-DEG). For the protein data, it extracts significant translation and protein degradation events (TRL/PROT-DEG). As such, PECA is capable of determining multiple levels of significant regulation per gene at each time point. We define a significant regulatory event if a change occurred in both replicates at a defined false discovery rate (FDR < 0.2), independent of the timing, but in the same direction, that is up- or downregulation (*Figure 1B*). All results presented here are robust to the use of stricter cutoffs and can be expanded to time-resolved analysis.

*Figure 1B* illustrates this workflow from measurements of concentrations and protein-RNA interactions to extraction of significant regulatory events for four types of processes. The examples are two stress-related genes, *HSPA5* (BiP/GRP78) and *HSP90B1*, the tryptophanyl-tRNA-synthetase Trp-tRS (*WARS*), and the DNA repair enzyme *RAD51*. *HSPA5* is a key chaperone of the ER stress response (*Lee, 2005*) and significantly upregulated in its transcription, translation, but also binding of non-ribosomal proteins to the untranslated regions (*Figure 1B*). As its protein concentration changed only little compared to the increase in transcription and translation, it is marked as downregulated by our analysis tool. Indeed, the *HSPA5* protein is known for its short half-life which delivers one explanation for the apparent counterbalancing regulation (*Shim et al., 2018*).

Several lines of evidence illustrate the quality of the data, the appropriate induction of protein misfolding stress, and the role of the hydrogen peroxide experiment as a control to identify ER stress specific changes. As expected (*Figure 1A*), the cancer cells exhibit low basal levels of ROS even without treatment. The ROS levels increase slightly upon tunicamycin exposure, likely due to the increased re-folding in the ER (*Figures 1A* and *2A*). Therefore, ER stress is indeed likely to elicit a weak oxidative stress response. However, even though we chose the lowest concentration known to induce oxidative stress in cancer cells (*Nakamura et al., 2003*), ROS levels increase much more when the cells are treated with hydrogen peroxide compared to tunicamycin, indicating the induction of a stronger response.

Tunicamycin treatment elicits the expected ER stress response in the cells as shown by an increase in phosphorylated *PERK* and phosphorylated eIF2$\alpha$ (*Figure 2B*). Note that, also as expected, the RNA and protein expression levels of these two stress markers do not change substantially (*Figure 2—figure supplement 1*). In comparison, the expression increase of markers of oxidative stress such as catalase, *SOD1*, and thioredoxin was modest in response to treatment with hydrogen peroxide (*Figure 2B*), consistent with the pre-existing ROS levels and the experimental setup. Further, when examining a set of 143 known UPR genes, many of the genes peak in their response at four hours after tunicamycin treatment (*Figure 2—figure supplement 2*). Finally, expression for many housekeeping genes remains largely unchanged, indicating cells are largely non-apoptotic (*Figure 2—figure supplement 3*).

The timeline of the ER stress response is further confirmed by the activation of UPR regulators, such as *XBP1*, *ATF4,* and *GADD34* (*PPP1R15A*)(*Figure 2C–E*). The non-canonical splicing of the fourth exon in *XBP1* results in a frame-shifted transcript (sXBP1) that produces the *XBP1* transcription factor; this is one of the earliest steps in the ER stress response (*Uemura et al., 2009*). While we observe some constitutive sXBP1 even in unstressed cells, the spliced isoform dominates by eight hours after tunicamycin treatment (*Figure 2C,D*). The spliced isoform is present in both the RT-qPCR and RNA-sequencing data, supporting our confidence in the resolution and quality of the large-scale dataset. Both *ATF4* and *GADD34* are known to escape global translation shutdown and are translationally upregulated via uORFs in the 5' UTR (*Lee et al., 2009*). GADD34 is responsible for the dephosphorylation of eIF2$\alpha$, making it essential for the reinitiation of translation necessary for cell survival and eventual apoptosis during chronic stress conditions (*Adler et al., 1999*; *Hollander et al., 1997*; *Marciniak et al., 2004*). Accordingly, we observe significant translation upregulation of *ATF4* and *GADD34* through an increase in ribosome footprints in the main open reading frame (*Figure 2—figure supplement 4*).

Finally, we validate our ability to extract significant regulatory events through examination of genes with known uORF-mediated translation increase during ER stress (*Figure 2E*). The genes include *ATF4*, *GADD34,* and *HSPA5* discussed above, as well as *ATF3*, *ATF5*, *DDIT3* (CHOP), *IBTK*, and *IFRD1*. Across all three replicates, we identify significant translation upregulation based on the increase in ribosome footprints compared to RNA levels (FDR < 0.2, TRL, *Figure 2E*). With the exception of *ATF5*, all genes also show concordant increases in RNA concentrations that are marked as significant regulatory events (FDR < 0.2, TRXP/RNA-DEG). *Figure 2—figure supplement 5* shows additional uORF- and IRES-regulated genes that serve as a negative control: these genes are not associated with ER stress and we did not not identify translation induction. The replicates are highly consistent.

## Intricate regulatory patterns emerge

*Figure 3* provides an overview of the complexity of the response to tunicamycin, the multiple routes of its regulation, and its relationship to the response to hydrogen peroxide treatment. As expected,

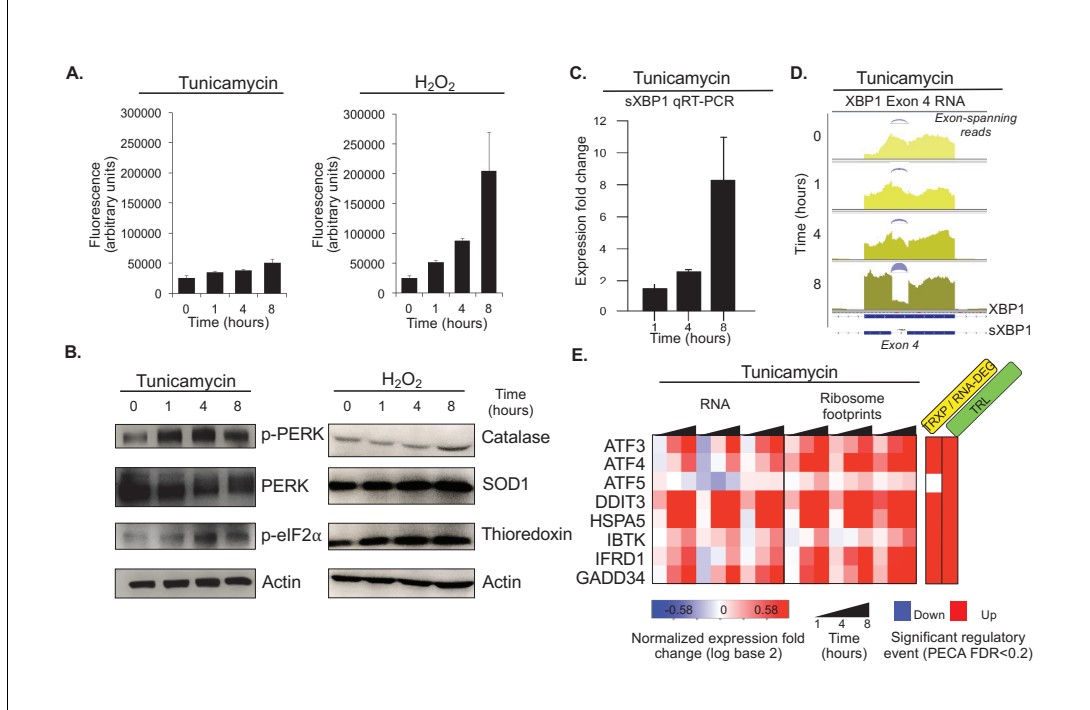

**Figure 2.** Tunicamycin elicits stress of the endoplasmic reticulum. (**A**) Reactive oxygen species (ROS) change in HeLa cells after tunicamycin and hydrogen peroxide (left and right, respectively) treatment, as measured for indicated time points. The cells show basal levels of ROS with minimal increase upon tunicamycin treatment, and substantial increase when treated with hydrogen peroxide, consistent with the relationships shown in *Figure 1A*. (**B**) The panels show phosphorylation levels of ER stress markers PERK and eIF2α increase after treatment with tunicamycin (left). Protein abundance increases for common markers of oxidative stress in hydrogen peroxide treated cells (right). (**C**) Splicing of *XBP1* (sXBP1) in tunicamycin treated cells increases, represented as mean fold change and standard error of the mean compared to the 0 hr time point. (**D**) Reads mapping to exon 4 of *XBP1* in the RNA-seq data indicate an increase in the spliced isoform in response to tunicamycin. Spliced reads spanning the 26 nucleotide intron and corresponding to sXBP1 are designated at each time point. Darker colors indicate later time points. (**E**) The heatmap shows normalized fold changes for all three replicates for RNA and ribosome footprints measurements of eight ER stress response genes regulated via upstream open reading frames. As the significance analysis indicates, the genes are indeed translationally upregulated. FDR - false discovery rate; p-PERK - phosphorylated PERK; p-eIF2α - phosphorylated eIF2α; TRXP - transcription; TRL - translation; RNA-DEG - RNA degradation; PROT-DEG - protein degradation.

DOI: https://doi.org/10.7554/eLife.39054.004

The following figure supplements are available for figure 2:

**Figure supplement 1.** Expression of PERK and eIF2α are largely unchanged.
DOI: https://doi.org/10.7554/eLife.39054.005

**Figure supplement 2.** UPR genes are upregulated during ER stress.
DOI: https://doi.org/10.7554/eLife.39054.006

**Figure supplement 3.** Housekeeping genes are largely unchanged.
DOI: https://doi.org/10.7554/eLife.39054.007

**Figure supplement 4.** Stress response genes are upregulated in transcription and translation.
DOI: https://doi.org/10.7554/eLife.39054.008

**Figure supplement 5.** Many uORF and IRES containing genes are translationally regulated.
DOI: https://doi.org/10.7554/eLife.39054.009

translation repression is the most frequent response to ER stress, affecting approximately one sixth of the 7,011 core genes (*Table 1*, N(TRL down)=1,189). However, many genes also increase in translation during ER stress (*Table 1*, N(TRL up)=746), including those with known mechanisms of translation induction via uORFs, as mentioned above. ER stress also elicits a large response at the transcript level, and in contrast to translation, we find twice as many transcriptionally up- than down-regulated genes (*Table 1*, N(TRXP; RNA-DEG up)=1,012 and N(TRXP; RNA-DEG down)=590). PECA results for the extended data show similar regulatory distributions (*Supplementary file 1*–

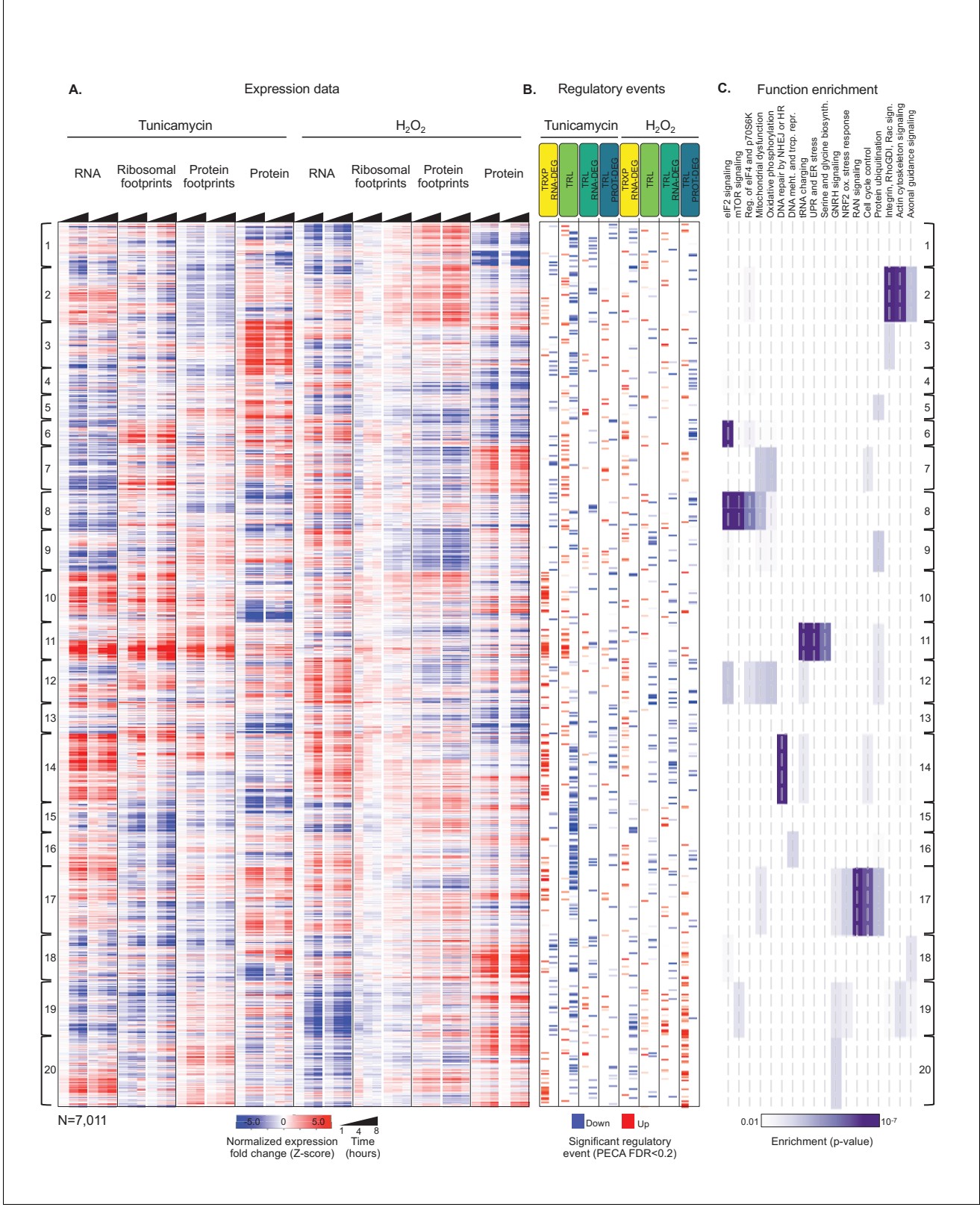

**Figure 3.** Integrated data reveal global and gene-specific regulation in response to misfolding stress. (A) The heatmap shows relative changes in RNA abundance, ribosome and non-ribosomal protein footprints, and protein abundance for 7,011 genes (rows) in duplicate after 1, 4 and 8 hr of tunicamycin or hydrogen peroxide treatment. Rows are sorted using complete-linkage hierarchical clustering and labeled according to their cluster number. We chose 20 clusters based on the 'elbow method' (*Figure 3—figure supplement 3*). (B) The heatmap indicates significant regulatory events

*Figure 3 continued on next page*

*Figure 3 continued*

for the genes at each regulatory level displayed along the top (false discovery rate <0.2 for both replicates). (C) The heatmap shows function pathways that are significantly enriched in the gene clusters (p-value<0.01). The order of the genes is the same across all three panels. FDR - false discovery rate; TRXP - transcription; TRL - translation; RNA-DEG - RNA degradation; PROT-DEG - protein degradation.

DOI: https://doi.org/10.7554/eLife.39054.011

The following figure supplements are available for figure 3:

**Figure supplement 1.** Replicates are consistent even prior to principal component analysis.
DOI: https://doi.org/10.7554/eLife.39054.012

**Figure supplement 2.** The results are robust to different significance thresholds.
DOI: https://doi.org/10.7554/eLife.39054.013

**Figure supplement 3.** Twenty clusters capture majority of biological variation.
DOI: https://doi.org/10.7554/eLife.39054.014

**Figure supplement 4.** *ATF4* and other upstream regulators target specific groups of genes.
DOI: https://doi.org/10.7554/eLife.39054.015

**Figure supplement 5.** Transmembrane proteins are often regulated at post-transcriptional levels.
DOI: https://doi.org/10.7554/eLife.39054.016

**Figure supplement 6.** Many cytosolic aminoacyl-tRNA synthetases increase in transcription and translation during ER stress.
DOI: https://doi.org/10.7554/eLife.39054.017

**Figure supplement 7.** Genes that are translationally up-regulated under ER stress often function in neurodegenerative diseases.
DOI: https://doi.org/10.7554/eLife.39054.018

*4*, *Figure 3—figure supplement 1*) and are robust to different significance thresholds (*Figure 3—figure supplement 2*).

To extract major trends in the data, we clustered the genes based on normalized expression changes across all data types and both stresses. We determined the appropriate cluster number through the 'elbow method' (*Figure 3—figure supplement 3*). Confirming successful induction of the ER stress response (*Figure 2*), many genes with similar expression changes in response to tunicamycin are enriched in UPR pathways (*Figure 3*, cluster 11, p-value<0.0001) and a quarter of these genes are upregulated concordantly in both transcription and translation (71 of 302 genes). Examples are genes discussed for *Figure 1* and *Figure 2* for example *HSPA5*, *GADD34*, and *HSP90B1*. This cluster also encompasses aminoacyl-tRNA synthetases (p<3×10−9) and serine biosynthesis enzymes (p<3×10−6) that are regulated in a manner similar to the UPR genes and are discussed later. Using an upstream regulator analysis (Ingenuity Pathway Analysis) we also confirm that cluster

**Table 1.** Many genes are regulated at different levels during stress.

The table summarizes the numbers of genes with significant regulatory events as defined by the PECA analysis. The events are split into stress-specific and shared events as discussed in *Figure 1*. Some shared events show significant function enrichment (false discovery rate <0.01, NCBI DAVID function annotation tool). * - Overlap significant with p-value<0.01 (hypergeometric test) TRXP - transcription; TRL - translation; RNA-DEG - RNA degradation; PROT-DEG - protein degradation

| Data type (Layer 2) | Data type (Regulation) | | Specific to ER stress (tunicamycin) | Specific to H$_2$O$_2$ treatment | Shared response | Function enrichment amongst shared genes |
|---|---|---|---|---|---|---|
| RNA | Transcription; RNA degradation (TRXP; RNA-DEG) | Up | 889 | 481 | 123* | DNA repair |
| | | Down | 529 | 417 | 61* | - |
| Ribosome footprints | Translation (TRL) | Up | 706 | 204 | 40* | - |
| | | Down | 1,116 | 249 | 73* | Cytoplasmic chaperones |
| Protein footprints | Translation; RNA degradation (TRL; RNA-DEG) | Up | 171 | 274 | 16* | - |
| | | Down | 371 | 359 | 40* | Cell adhesion |
| Protein | Translation; protein degradation (TRL; PROT-DEG) | Up | 235 | 614 | 24 | - |
| | | Down | 358 | 492 | 32 | - |

DOI: https://doi.org/10.7554/eLife.39054.010

11 is enriched for genes that are themselves targets of established UPR regulators (e.g. *ATF4* and *ATF6*) (*Figure 3—figure supplement 4*).

We observe a significant increase in protein expression during ER stress for a set of genes significantly enriched in the integrin signalling pathway (*Figure 3*, cluster 3, p<0.0001, *Figure 3—figure supplement 5*). Integrins are transmembrane proteins that are synthesized and processed in the ER (*Tiwari et al., 2011*), and indeed, cluster three is generally enriched in transmembrane proteins (*Figures 3* and 161 of 377 genes, p<0.0001). Many of these genes decrease in translation during ER stress, but show stable or increasing protein abundance (*Figure 3—figure supplement 5*), suggesting that as the ER becomes incapable of synthesizing new transmembrane proteins during stress, those already present within membranes may be stabilized to preserve cellular function until synthesis of new proteins is able to resume.

## Many genes are regulated in multiple ways

Next, we examined the interplay between different regulatory pathways that can affect the same gene in its response to ER stress. This analysis is illustrated at the example of enzymes from four DNA repair pathways, which include *RAD51* mentioned above (*Figure 1B*). Like *RAD51*, many of the DNA repair genes, for example *NBN* and *MRE11A* functioning in homologous recombination, are significantly induced in their transcript levels, but also significantly repressed in their translation in response to ER stress, counterbalancing the effect of transcription (*Figure 4A,B*; TRXP; RNA-DEG vs. TRL).

We examined these cases, in which genes are regulated in more than one way, more closely. In some cases, the events were discordant, such is the case for the DNA repair genes: the cell appears to regulate two processes in opposing directions, that is increasing transcription but decreasing translation (*Figure 4B*). Discordance can be implemented in different ways at the molecular level. For example, some genes, like *MRE11A* and *NBN*, show decreases in the abundance of translating ribosomes under ER stress and also a decrease in protein concentration, opposing the increase in transcript numbers (*Figure 4A*). Other genes, like *RAD50, BRCA1,* and *BRCA2*, show an increase in both RNA abundance and ribosome footprints; however, the increase in ribosome footprints is small, resulting in an overall decrease in the translation for these transcripts compared to their transcription increase. When examining all 7,011 genes, discordant regulation frequently involves post-transcriptional regulation opposing what was initiated by transcriptional changes (*Figure 4C*), suggesting that counterbalancing regulation might be more common than previously thought.

In comparison, concordance often occurs amongst different post-transcriptional pathways, that is between events extracted from ribosome footprinting, protein occupancy profiling, and protein expression data, but not transcription. This effect is perhaps due to translation being common across the post-transcriptional categories. However, we also find a modestly significant amount of concordant regulation between transcription and translation in response to ER stress, including among many UPR genes mentioned above (*Figure 3*, cluster 11; *Figure 4C*, p<0.1, hypergeometric test).

Next, we examine the timing of the different regulatory processes. The analysis is again illustrated for the example of DNA repair genes, specifically those with transcription up- and translation downregulation (*Figure 4D*). To do so, we averaged for each gene the time points at which significant regulation was observed across replicates and plot the frequency distributions. The transcription of DNA repair genes is induced early after treatment, but translation is repressed throughout the mid-to-late phases of stress. This delay between the two events is perhaps due to a rapid increase in mRNA abundance upon the sensing of stress that subsequently outpaces the availability of ribosomes.

*Figure 4E* takes this temporal analysis to all 7,011 genes. The resulting distributions for up- and downregulation are surprisingly symmetrical. During ER stress, most of the transcripts change early at one hour of tunicamycin treatment. Some genes, such as those of the UPR, also change early in translation, others change later (*Figure 4E*). The late response is more pronounced for the protein expression data than for the ribosome footprinting data. This delay in post-transcriptional regulation appears to be different from the canonical translation shutdown and perhaps reflects alternative mechanisms that recover protein synthesis from stress.

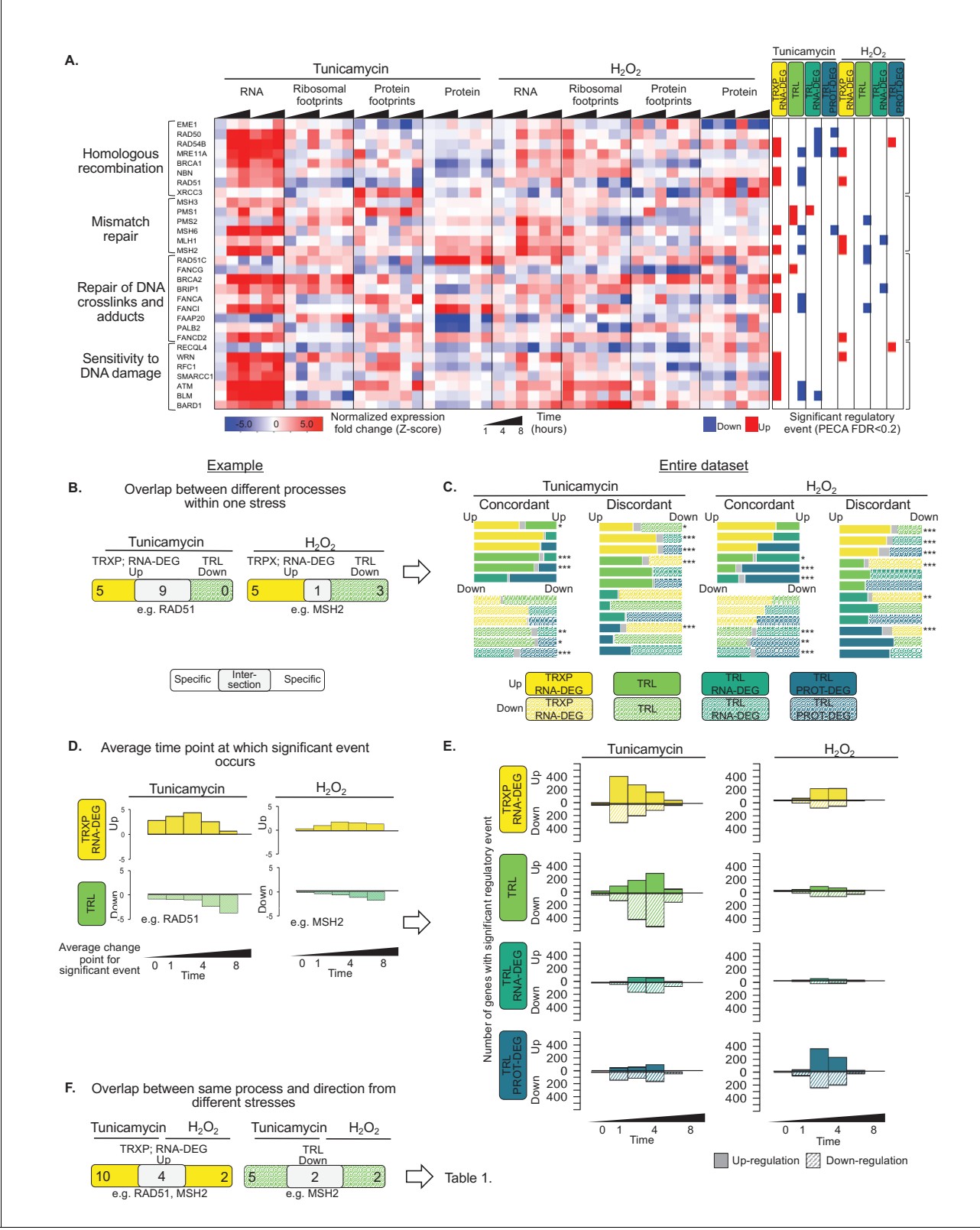

**Figure 4.** Gene expression is differentially regulated across processes, time, and stress types. (**A**) RNA abundance, ribosome and non-ribosomal protein footprints, and protein abundance for genes involved in selected DNA repair pathways change in distinct ways, with significant regulation at each regulatory level as determined by PECA (false discovery rate <0.2 in both replicates). *Figure 4—figure supplement 1* shows the data for all DNA repair genes. (**B**) The Venn diagrams illustrate the overlap (intersection) between genes regulated at different levels at the example of the 30 DNA repair

*Figure 4 continued on next page*

*Figure 4 continued*

genes shown in (A). Concordant or discordant regulation can be extracted through comparison of the number of genes with significant regulatory events affecting gene expression in the same or different directions, respectively. These genes are in the intersection displayed in grey. *RAD51* and *MSH2* are two examples for such discordantly regulated genes in which transcription/RNA degradation is significantly upregulated, but translation is significantly downregulated. (C) Venn diagrams show concordant and discordant regulation for the entire dataset of 7,011 genes. Bar sizes represent relative gene numbers within each group, and the intersection in grey. The intersection/overlap contains the genes with concordant or discordant regulation: the genes show significant regulation for two processes. We used the hypergeometric test to determine if a significant number of genes was regulated by two processes and placed in the intersection (*p-value<0.1,**p-value<1×10−3,***p-value<1×10−6). For example, both tunicamycin and hydrogen peroxide treatment show a significant number of genes with discordant regulation in which transcription is upregulated, but translation decreases. These genes are enriched in DNA repair genes which are discussed in the text and shown panel (A) and (B, D) We deconvoluted the regulatory events identified for DNA repair genes with respect to time: graphs indicate when the event occurred during the 8 hr experiment, as averaged over two replicates. A positive y-axis shows upregulated genes (transcription), while negative y-axis represents downregulated genes (translation). (E) The panels show the time-dependent occurrence of significant events for the entire dataset shown in *Figure 3*. The panels show from top to bottom: TRXP/RNA-DEG; TRL; TRL/RNA-DEG; TRL/PROT-DEG. (F) The Venn diagrams illustrate at the example of the DNA repair genes the analysis of the relationship between the two stresses. *RAD51* and *MSH2* are both transcriptionally upregulated in response to either treatments, while *MSH2* is decreases in translation in both treatments. The results for the entire data are discussed in *Table 1*. FDR - false discovery rate; TRXP - transcription; TRL - translation; RNA-DEG - RNA degradation; PROT-DEG - protein degradation.

DOI: https://doi.org/10.7554/eLife.39054.019

The following figure supplement is available for figure 4:

**Figure supplement 1.** DNA repair pathways and their regulation during stress.

DOI: https://doi.org/10.7554/eLife.39054.020

## The shared stress response involves few genes but highly similar patterns

Next, we attempted to disentangle the ER stress response from a general stress response and the response due to an increase in reactive oxygen species in the cell. Due to the interconnected pathways (*Figure 1A*) and the basal ROS levels in cancer cells (*Figure 2A*), we chose a hydrogen peroxide concentration at the lower range of what is known to elicit oxidative stress in HeLa cells (*Nakamura et al., 2003*).

We find that the response to hydrogen peroxide treatment is qualitatively very similar to the response to ER stress (*Table 1*), but is overall much less pronounced (*Figure 3*). Only about a tenth of significant regulatory events in each category is shared across the two treatments (*Table 1*). Despite these small fractions, the number of shared genes is significant for most processes (p-value<0.05, hypergeometric test). In contrast to events specific to ER stress, which are dominated by translation, the shared response is dominated by up- and downregulation of transcription and RNA degradation (*Table 1*, TRL vs. TRXP; RNA-DEG).

Hydrogen peroxide treatment elicited similar patterns with respect to concordant and discordant regulation as compared to ER stress (*Figure 4C*) and similar temporal distributions (*Figure 4E*). The genes of the shared stress response, that is genes whose transcription is induced in response to both ER and oxidative stress, are enriched in DNA repair pathways discussed above as examples of discordant regulation (*Table 1*, p<0.01). Similar to ER stress, many DNA repair enzymes increase in their transcription in response to hydrogen peroxide treatment, but decrease in translation. Many of these genes belong to cluster 14 in *Figure 3*, and regulatory patterns for a subset are shown in *Figure 4A*, with the extended set in *Figure 4—figure supplement 1*.

## Proteins from different subcellular localizations show different modes of regulation

The core set of 7,011 genes examined here is representative of the cytosol, nucleus, and various organelles, as assigned from published data (*Figure 5A*)(*Itzhak et al., 2016*). Consistent with the above results, we observe much upregulation of RNA levels (TRXP/RNA-DEG) upon tunicamycin treatment for genes whose proteins reside in the ER (p<0.009; *Figure 5B*). These genes include many known UPR targets that relieve the burden of accumulating misfolded proteins.

Proteins localized to plasma membranes are marked by significantly reduced binding of non-ribosomal proteins to the 3' UTR of the corresponding mRNAs (TRL/RNA-DEG; p<0.002; *Figure 5B*). Upon closer examination, we find that many of these genes include transmembrane proteins that

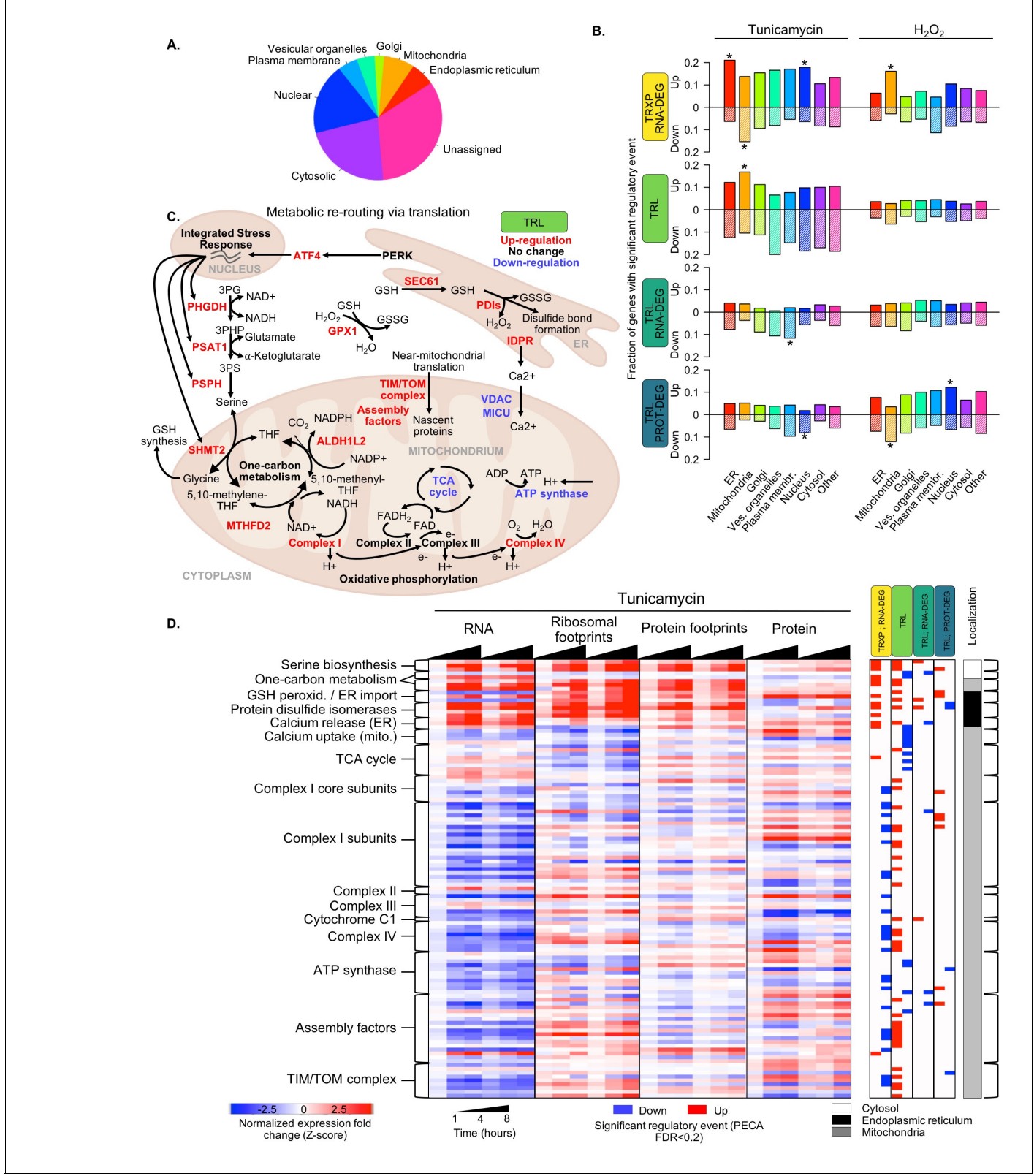

**Figure 5.** Translation regulation supports re-routing in energy metabolism. (**A**) The 7,011 proteins in the core dataset map to different subcellular localizations (*Itzhak et al., 2016*). (**B**) Panels indicate the fraction of organelle-specific genes that are regulated within the different processes responding to the two types of stress. A positive y-axis represents upregulated genes, while negative y-axis represents downregulated genes. Fisher's exact tests were used to assess whether up- or down-regulated genes were independent of each organelle. Significant differences in the distributions

*Figure 5 continued on next page*

*Figure 5 continued*

compared to all genes are indicated by * (adjusted p-value<0.05). (**C**) Significant changes in translation of different pathways suggests rerouting of energy production to involve one-carbon metabolism. The diagram illustrates the simplified relationships between genes in panel (**D**). Translation up- or down-regulation of genes and pathways is indicated by color. (**D**) The heatmap shows the data collected for mitochondrial and related genes discussed in panel (**C**), responding to ER stress. *Figure 5—figure supplement 1* summarizes the translation changes for genes from oxidative phosphorylation, the ATP synthase complex, and the tricarboxylic acid cycle. FDR - false discovery rate; TCA - tricarboxylic acid; TRXP - transcription; TRL - translation; RNA-DEG - RNA degradation; PROT-DEG - protein degradation.

DOI: https://doi.org/10.7554/eLife.39054.021

The following figure supplement is available for figure 5:

**Figure supplement 1.** Translation changes of oxidative phosphorylation and related genes.

DOI: https://doi.org/10.7554/eLife.39054.022

require processing in the ER and therefore particularly burden the organelle during stress. These proteins often undergo N-linked glycosylation (p<0.0001), the precise process blocked by tunicamycin that leads to misfolding. Therefore, it is tempting to speculate that the reduction in binding of non-ribosomal proteins to the mRNAs of these genes acts to repress the translation of these problematic mRNAs or trigger their degradation. When examining regulation of these transmembrane proteins more closely, we noticed that the former explanation is more likely than the latter: transcript levels of the proteins do not change or increase indicating stabilization of the mRNA, while ribosome binding decreases indicating reduced translation (*Figure 3—figure supplement 5*).

Proteins localized to mitochondria encompass the only category of genes significantly upregulated in translation in response to ER stress (TRL, p<0.0003; *Figure 5D*). A similar, although non-significant trend is also observed at the protein level (TRL/PROT-DEG). *Figure 5* and the section below discuss these genes in more detail.

## Translation regulation modulates energy metabolism during ER stress

Closer examination of genes whose proteins localize to mitochondria indicates a shift in energy metabolism in response to ER stress from the tricarboxylic acid (TCA) cycle to one-carbon metabolism to feed oxidative phosphorylation. Translationally upregulated genes are enriched in subunits of Complexes I-IV from the oxidative phosphorylation pathway (*Figure 5C,D*, *Figure 5—figure supplement 1*). While the genes' mRNA levels often decrease during ER stress, ribosome binding and therefore translation increases significantly (p<0.0001, *Figure 5C,D*). In contrast, translation is largely unchanged among subunits of the ATP synthase and decreases among enzymes involved in the TCA cycle with the exception of *ACO2* (*Figure 5C,D*, *Figure 5—figure supplement 1*.

Examination of the 5' UTR of the genes for Complex I-IV reveals a sequence element that might explain their differential translation increase during ER stress. Translation of mitochondrial genes is often regulated by the TISU element, the Translation Initiator of Short 5' UTR (*Elfakess and Dikstein, 2008*; *Elfakess et al., 2011*). The TISU element is also part of mTOR-sensitive mRNAs, which in turn confers resistance to translation inhibition under energy deprivation (*Gandin et al., 2016*; *Sinvani et al., 2015*). Therefore, we hypothesize that TISU elements may play a role in the translation regulation of mitochondrial genes during ER stress, as this condition challenges the cell's energy balance. Indeed, we find the TISU element (SAAS**AUG**GCGGC) highly enriched among the translation initiation sites of Complex I-IV genes (E-value = $1.7 \times 10^{-8}$), but not amongst genes for ATP synthase subunits or TCA cycle enzymes (E-value = 0.03 and E-value = 0.1, respectively).

We also observe increased translation for other mitochondrial components that are related to either oxidative phosphorylation or the ER stress response, including one-carbon metabolism, TIM/TOM complexes, respiratory-chain assembly factors, and calcium uptake (*Figure 5C,D*). Of particular interest is the subset of one-carbon metabolism genes whose proteins localize to mitochondria (*SHMT2*, *MTHFD2*, and *ALDH1L2*). While the TCA cycle generally supplies NADH that feeds into the oxidative phosphorylation pathway to generate ATP, production of mitochondrial NADH can also be achieved through one-carbon metabolism, particularly under conditions of stress (*Ducker and Rabinowitz, 2017*). Translation increase of one-carbon metabolism genes with simultaneous translation repression of TCA cycle genes, as we observe in our data (*Figure 5C,D*, *Figure 5—figure supplement 1*), suggests a rerouting of NADH production.

Our data indicates an important role of serine metabolism in this process. NADH production via one-carbon metabolism is driven by serine, which can be generated through the upstream serine biosynthesis pathway in the cytosol (*PHGDH, PSAT1, PSPH*)(*Figure 5C,D*). While enzymes of serine biosynthesis are known to be transcriptionally induced in various stresses (*Zhao et al., 2016*; *Zhou et al., 2017*), we find additional upregulation of translation for these genes (*Figure 5C,D*). We also find such concordant increase in transcription and translation for the mitochondrial one-carbon metabolism enzymes, but not for their cytosolic homologs, supporting our interpretation.

Finally, components of the TIM/TOM complexes and respiratory-chain assembly factors, also exhibit discordant regulation during ER stress, similar to Complex I-IV: they decrease in mRNA abundance but increase in translation (*Figure 5C,D*). *TIMM17A* represents an exception as it significantly decreases at the protein level, consistent with previous findings (*Rainbolt et al., 2013*). Co-translational import and proper assembly of oxidative phosphorylation complexes is orchestrated by TIM/TOM and complex specific assembly factors, and dysregulation of this process leads to complex defects implicated in many human diseases (*Calvo et al., 2010*; *Mick et al., 2012*; *Vogel et al., 2005*; *Weraarpachai et al., 2009*; *Heide et al., 2012*; *Mckenzie and Ryan, 2010*). Therefore, this shared regulatory pattern may reflect a coordinated response between the oxidative phosphorylation complexes and their assembly machinery, which work together to adjust mitochondrial metabolism when the cell is burdened by misfolded proteins in the ER.

## Alternative splicing of aminoacyl-tRNA synthetases exemplifies the variety of stress-induced regulation

As another case study, we examined a functional family of genes in the same cluster as the UPR and serine biosynthesis genes, which also shows substantial transcription and translation upregulation: aminoacyl-tRNA synthetases (*Figure 3*, cluster 11; *Figure 3—figure supplement 6*). Post-transcriptional regulation of these enzymes in response to stress has been observed before without an explanation or a specific link to ER stress (*Cheng et al., 2016*; *Ventoso et al., 2012*). As these enzymes are known for their additional functions beyond aminoacylation of tRNAs (*Guo and Schimmel, 2013*) and have been observed to undergo alternative splicing related to such 'moonlighting' (*Lo et al., 2014*), we hypothesized that stress-dependent expression of alternative transcript variants might explain some of discrepancy between changes in ribosome binding and the corresponding protein levels observed in our data.

Indeed, we find evidence for ~80 alternative splicing events amongst the 20 cytosolic aminoacyl-tRNA synthetases, where the RNA reads on exon-exon junctions indicate the expression of a second transcript variant (*Figure 6A*). The distribution of these events is similar to what has been observed across the entire human genome, with exon skipping being the most frequent splicing event (*Sammeth et al., 2008*). A small number of these alternative splicing events changes in response to ER or oxidative stress consistently across the three replicates; these examples indicate potential stress-dependent splicing. One such example is the tryptophanyl-tRNA synthetase *WARS* (*Figures 1* and *6B-E*). Other examples are shown in *Figure 6—figure supplement 1* and *Supplementary file 6*.

*WARS* has five previously identified transcript variants in total (*Figure 6B,E*). Two transcript variants exclude exon II, leading to the production of a shorter protein called mini-TrpRS which misses the N-terminal protein interaction domain (*Tolstrup et al., 1995*; *Wakasugi et al., 2002*); mini-TrpRS is non-catalytic. For both the RNA and the ribosome footprint data, reads signifying skipping of exon II and production of mini-TrpRS increased at one hour after tunicamycin induced ER stress (*Figure 6C*). This trend is confirmed by a decrease in the abundance of the full-length protein as seen in the proteomics data (*Figure 6—figure supplement 2*). As mini-TrpRS is known for its anti-proliferative function and its role in angiogenesis (*Wakasugi et al., 2002*; *Nakamoto et al., 2016*), the cell might produce the short variant in response to stress to reduce proliferation.

Excitingly, our data describes another splice event for *WARS* in response to ER stress, letting us hypothesize on a possible new regulatory mechanism affecting the production of mini-TrpRS. Exons -Ia and Ia in the 5' UTR of *WARS* are mutually exclusive (*Figure 6B*), and we find a substantial increase in the inclusion of exon -Ia at one hour after stress treatment, correlating with the exclusion of exon II (*Figure 6D*). Indeed, when combining the data for the two splicing events, we observe a significant bias towards exclusion of exon II and production of mini-TrpRS if exon -Ia instead of Ia is used (p-value<0.001, *Figure 6—figure supplement 2*). We hypothesize that this splice event in the UTR influences skipping of exon II.

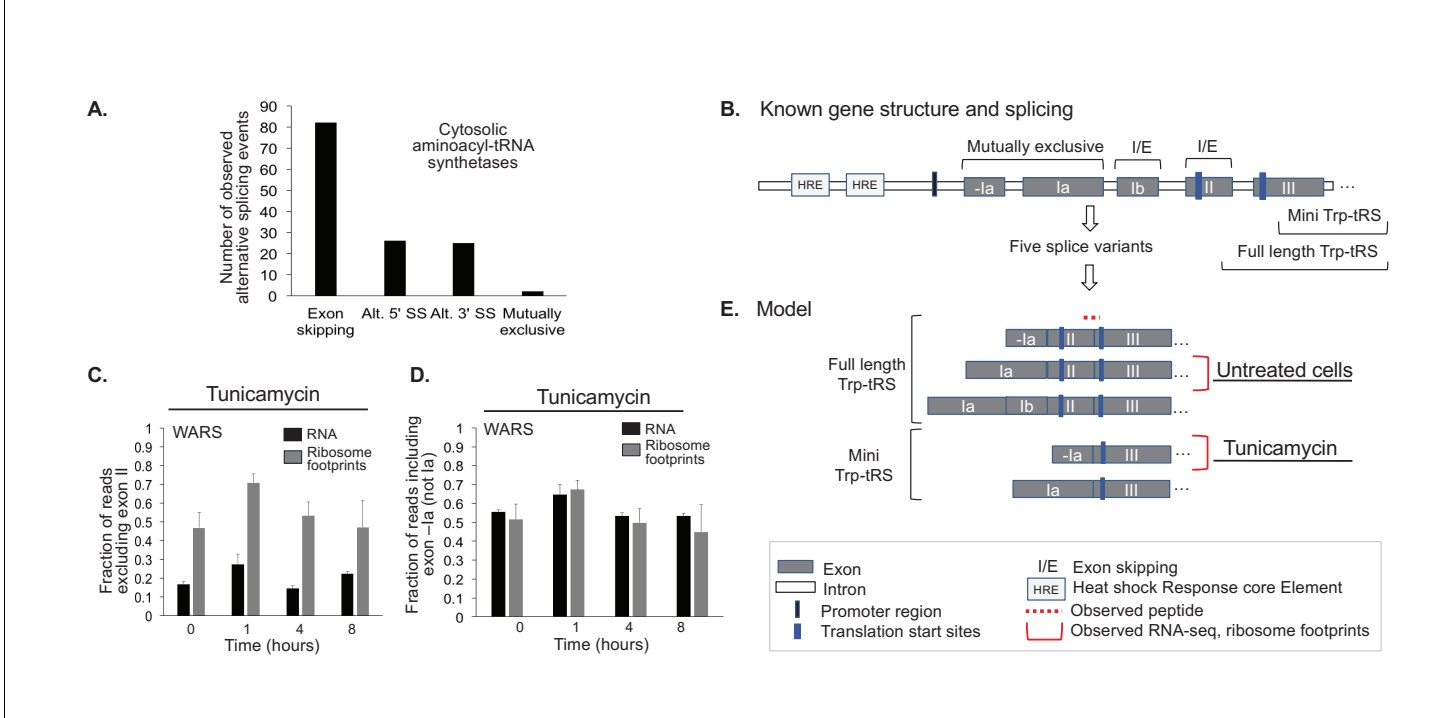

**Figure 6.** Aminoacyl-tRNA synthetases show evidence for alternative splicing under stress. (**A**) Aminoacyl-tRNA synthetase genes are extensively spliced. We detected a total of 82 exon skipping events, 26 alternative 5' splice sites (Alt. 5' SS), 25 alternative 3' splice sites (Alt. 3' SS), and two mutually exclusive exons amongst cytosolic aminoacyl-tRNA synthetases. (**B**) Alternative splicing for some genes correlates with the stress response, as illustrated for tryptophanyl-tRNA-synthetase responding to ER stress (Trp-tRS, *WARS*). The diagram is not drawn to scale. The long and short protein isoforms of Trp-tRS are known to be encoded by five transcript variants, each with a different 5'UTR that arises through alternative splicing. When exon II is spliced in, translation starts on exon II, leading to a full-length protein isoform. When exon II is skipped, translation starts on exon III, resulting in the truncated mini-TrpRS isoform. (**C**) Skipping of Trp-tRS' exon II, which results in the mini-TrpRS isoform, increases after one hour of tunicamycin treatment in both the RNA and ribosome footprinting data. Expression of exon II decreases again after the first hour.(**D**) *WARS*' exons -Ia and Ia are mutually exclusive. The fraction of reads including exon -Ia increases at one hour after tunicamycin treatment similar to the skipping of exon II shown in panel (**C**). The exon II skipping event is significantly more likely to occur when transcription starts on exon -Ia (p-value<0.01, *Figure 6—figure supplement 2*). (**E**) Our data suggests two interdependent splicing events that promote production of the short isoform mini-Trp-tRS in response to tunicamycin treatment. The peptide AGNASKDEIDSAVK spans the exon II/III junction and is only present in the full-length isoform,its expression is shown in *Figure 6—figure supplement 2*.

DOI: https://doi.org/10.7554/eLife.39054.023

The following figure supplements are available for figure 6:

**Figure supplement 1.** Alternative splicing for leucyl-tRNA synthetase (*LARS*).

DOI: https://doi.org/10.7554/eLife.39054.024

**Figure supplement 2.** Extended analysis for splicing events in tryptophanyl-tRNA synthetase (*WARS*).

DOI: https://doi.org/10.7554/eLife.39054.025

## Ribosomes and post-transcriptional regulators bind to or around conserved RNA secondary structures

Finally, we provide new hypotheses for mechanisms and regulators underlying post-transcriptional regulation through stress-dependent binding of proteins to conserved secondary structures in the mRNAs' untranslated regions (UTR)(*Figure 7*). The importance of RNA secondary structures in the UTR has been increasingly recognized, especially in response to stimuli (*Leppek et al., 2017*; *Mustoe et al., 2018*; *Wu and Bartel, 2017*). Structures in the 5' UTR can form Internal Ribosomal Entry Sites (IRES), which bind to ribosomes and allow translation in a cap-independent manner (*Holcik et al., 1999*; *Macejak and Sarnow, 1991*), and inhibitory elements that prevent the bypass of ribosomes leading to reduced translation (*Xue et al., 2015*; *Xue and Barna, 2015*). In the 3' UTR, secondary structures can mitigate binding of regulators that impact stability of mRNAs, translation efficiency, and mRNA localization (*Chang et al., 2010*; *Mignone et al., 2002*). As our dataset

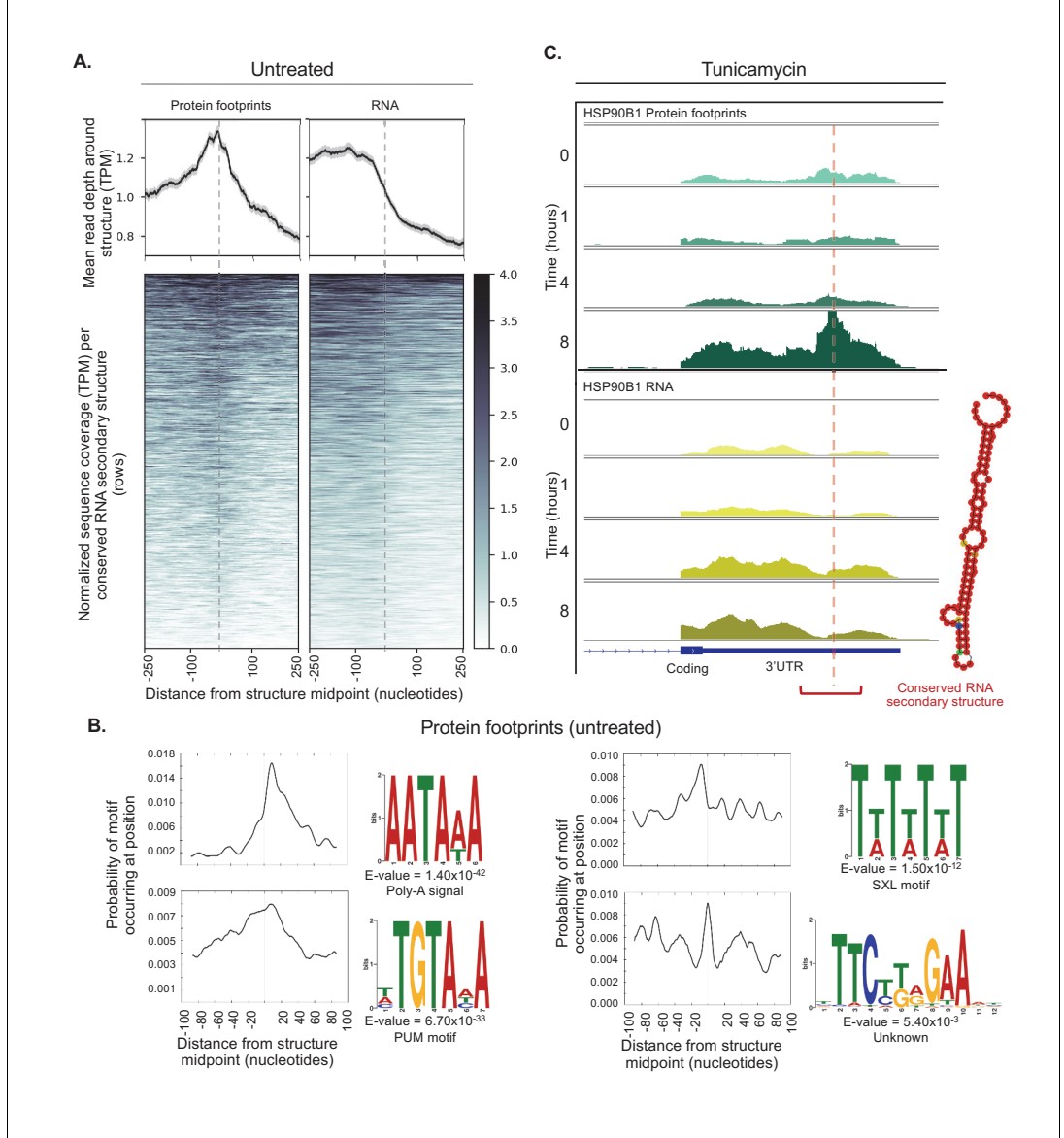

**Figure 7.** Conserved RNA secondary structures are enriched in protein footprints. (**A**) Top panels show mean read depth (transcripts per million) of conserved RNA secondary structures in the 3' UTR in a window ±250 nucleotides for protein footprints and RNA, respectively. Data was collected from untreated cells. Bottom panels show coverage for individual secondary structures over the 500 nucleotide window: each row corresponds to one secondary structure. Dashed lines indicate the midpoint of each secondary structure. (**B**) The sequence logos illustrate motifs enriched in sequences corresponding protein-bound RNA secondary structures. Probabilities describe change of the motif occuring in a ± 100 nucleotide window around the structure midpoints. PUM (Pumilio) is known to bind to single stranded RNA. SXL (Sex-lethal) does not have an ortholog in human, but the human RNA-binding protein *ELAVL1* (HuR) has a similar binding motif. (**C**) The profiles show protein footprints (top) and RNA read coverage (bottom) for the 3' UTR of the *HSP90B1* chaperone, for one replicate after 0, 1, 4, or 8 hr of tunicamycin treatment. Dashed lines indicates the midpoint of the predicted RNA secondary structure; the structure location is indicated by the red bracket. The predicted RNA secondary structure on the right was generated using minimal free energy while incorporating structural reactivities acquired from publically available probing-based experimental data.

DOI: https://doi.org/10.7554/eLife.39054.026

The following figure supplements are available for figure 7:

**Figure supplement 1.** Predicted conserved RNA secondary structures in the 3'UTR have a higher structure score than random 3'UTR control sequences.

DOI: https://doi.org/10.7554/eLife.39054.027

**Figure supplement 2.** Predicted conserved secondary structures in the 5'UTR of ferritin heavy and light chain mRNA (*FTH1* and *FTL*) display ribosomal occupancy surrounding structure midpoints.

DOI: https://doi.org/10.7554/eLife.39054.028

*Figure 7 continued on next page*

*Figure 7 continued*

**Figure supplement 3.** Predicted conserved secondary structures in the 3'UTR of ER stress response mRNAs (*HSPA5* and *CALR*) show stress dependent protein occupancy.

DOI: https://doi.org/10.7554/eLife.39054.029

includes RNA footprinting data for both ribosomes and non-ribosomal proteins, we sought to identify local binding events in the UTR that may reflect stress-dependent interactions by focusing on a set of regions with predicted RNA secondary structures (*Parker et al., 2011*).

These elements were identified based on similarity of sequence and predicted secondary structure across vertebrate genomes (*Parker et al., 2011*), however the majority of these have not been experimentally validated or linked to biological functions. Therefore, to validate these predictions, we computed an aggregate structure score for the 5,504 predicted conserved structures that mapped to the 3' UTR in our dataset and compared this value to 5,504 random 3' UTR sequences. The aggregate structure score was derived from the Structure Surfer database (*Berkowitz et al., 2016*) and included experimental data from Parallel Analysis of RNA Structure (PARS)(*Wan et al., 2014*)(*Figure 7—figure supplement 1*). The predicted structures used for the analyses below show a higher average score for the 200 base pair region surrounding the structure midpoint compared to the random sequences, suggesting these conserved secondary structures are enriched for truly structured regions. This enrichment is even stronger for the subset of sequences which contained protein footprints observed in our data (*Figure 7—figure supplement 1*), confirming both the validity of our experimental methods and the prediction of RNA secondary structures.

Another confirmation of the validity of the predicted structures lies in identification of *bona fide* regulatory elements, for example the iron response element. Two predicted RNA structures with the highest abundance of ribosome footprints are located within the 5' UTR of the ferritin heavy and light chain transcripts, *FTH1* and *FTL* (*Figure 7—figure supplement 2*). These structures correspond to iron response elements that interact with ribosomes to regulate translation according to cellular iron levels (*Aziz and Munro, 1987*; *Hentze et al., 1987a*; *Hentze et al., 1987b*). In total, we find transcript information for 302 secondary structures predicted in 5' UTRs (*Table 2*, *Supplementary file 7*). Of these, two-thirds (190) have ribosome footprints mapping to a window 100 nucleotides upstream or downstream of the structure midpoint. It is tempting to speculate that ribosomes might be stalled at these predicted structures.

In addition, we observe a large number of genes with predicted secondary structures in the 3' UTR, and for more than two-thirds of them non-ribosomal protein footprints map to a 200

**Table 2.** Conserved RNA secondary structures accumulate ribosome and non-ribosomal protein footprints.

The table summarizes the combined analysis of conserved RNA secondary structures as predicted by reference (*Parker et al., 2011*) and the expression data. As expected, ribosome footprints accumulate around secondary structures in the 5'UTR of the genes more than in the 3'UTR, while this is the opposite for protein footprints. Listed numbers show data for all structures across the extended dataset (*Supplementary file 1–3*); the number of unique genes with predicted structures are indicated in parentheses.

| | 5' Untranslated region | 3' Untranslated region |
| --- | --- | --- |
| Conserved RNA secondary structures predicted in human genes | 760 (626) | 5,504 (2,789) |
| Conserved RNA secondary structures with RNA expression | 302 (277) | 2,697 (1,633) |
| Conserved RNA secondary structures with ribosome footprints | 190 (183) | 154 (151) |
| Conserved RNA secondary structures with protein footprints | 89 (81) | 1,808 (1,166) |

DOI: https://doi.org/10.7554/eLife.39054.030

nucleotide window surrounding the structure (1,166 of 1,633 transcribed genes with structures, *Table 2*). When averaging across all transcribed 3' UTR structures in untreated data (0 hr time point), we find that protein footprints are distributed symmetrically around the structure midpoint, suggesting proteins can directly bind these secondary structures (*Figure 7A*, *Supplementary file 7*). While the identity of putative post-transcriptional regulators at these sites remains to be determined, we observe a clear enrichment of footprints of non-ribosomal proteins around the structures. RNA structures are predominantly located near the 3' ends of transcripts, which results in declining read coverage downstream of the structures (*Figure 7A*, right panel, RNA).

The presence of sequence motifs in close proximity to protein-bound RNA secondary structures further supports the possible binding of regulatory factors, that is RNA-binding proteins (*Figure 7B*). The most significant motif corresponds to the poly(A) signal site (AAUAAA, E-value = $1.40 \times 10^{-42}$). In our data this motif is most frequently 10 to 20 nucleotides downstream of protein-bound RNA structures, in line with recent work showing in vivo folded 3'-end structures are often located near poly-A signals (*Wu and Bartel, 2017*)(*Figure 7B*).

Other observed motifs include predicted binding sites for Pumilio and Sex-Lethal (PUM and SXL), as well as a hairpin without any predicted motif or known RNA-binding partner (*Figure 7B*). As SXL is an RNA-binding protein in *Drosophila* involved in sex determination without a known homolog in human, the motif observed in our data is therefore likely occupied by a protein with similar binding preferences, such as *ELAVL1* which is known to bind to AU-rich elements in the 3' UTR (*Ma et al., 1996*; *Wang et al., 2013*). Footprints for each of these motifs have a different spatial arrangement around the midpoint of structures. Pumilio RNA-binding sites display a broad distribution roughly centered around the structure (*Figure 7B*), while SXL motifs have a distinct spacing pattern, with the highest frequency 8 to 10 nucleotides upstream of structure midpoints and additional sites roughly 10 to 15 nucleotides apart. Finally, the hairpin is often the center of the RNA structure itself, but also appears 30 to 40 nucleotides away from the structure (*Figure 7B*).

While not all binding events surrounding predicted secondary structures are expected to be stress responsive, our data does contain examples in which ER stress alters the occupancy of 5' and 3' UTR structured regions, either by ribosomes or non-ribosomal proteins, respectively. We identified these examples by comparing the local coverage of structures at 0 and 8 hr time points and extracted significant changes (*Supplementary file 7*). We observe an increase in local ribosome footprints for a single 5' UTR structure located within in the *TSC22D3* transcript (adjusted p-value=0.031). This transcript encodes the glucocorticoid-induced leucine zipper protein (*GILZ*), which has been shown to act as a pro-survival factor during ER stress (*André et al., 2016*).

Further, we detect significant increases in local protein binding for five conserved structures predicted in 3' UTRs (adjusted p<0.01). Remarkably, these structures are all within genes that have an established role in the UPR, including three molecular chaperones that act as critical sensors of misfolded proteins: *HSP90B1*, *HSPA5*, and *CALR* (*Lee, 2005*; *Huang et al., 2014*; *Eletto et al., 2010*; *Mungrue et al., 2009*; *Ellgaard and Helenius, 2003*). We observe a clear increase in protein binding over these conserved secondary structures after eight hours of tunicamycin treatment, independent of changes in RNA reads mapping to the transcripts, suggesting an important role in their regulation during stress (*Figure 7C*, *Figure 7—figure supplement 3*).

## Discussion

### Both transcriptional and post-transcriptional regulation govern the stress response

We present a high-quality, information-rich resource that provides new insights at multiple levels at which cancer cells respond to either ER or oxidative stress. In measuring not only RNA and protein abundances, but also ribosome and other protein footprints along mRNAs, we capture the adaptive landscape of various stress response pathways during the first eight hours of stress. We extract hundreds of genes with significant changes in transcription, translation, and degradation at individual time points and isolate regulatory signatures that have been hitherto unknown or underappreciated.

Translation regulation is a key component of the mammalian UPR, as translational shutdown is one of the earliest reactions to ER stress. However the extent to which genes can evade this global inhibition and the ways in which they do is an active area of investigation. We identify many genes

that significantly increase in translation during ER stress (*Table 1*, *Figure 3*), expanding upon the few well-known cases previously described (*Vattem and Wek, 2004*; *Lee et al., 2009*; *Palam et al., 2011*; *Zhao et al., 2010*; *Zhou et al., 2008*; *Baird et al., 2014*), as well as what has been suggested by genome-wide profiling experiments (*Baird et al., 2014*; *Guan et al., 2017*; *Ventoso et al., 2012*). Importantly, the increase in translation for many of these genes is independent of transcription. Through comparison with data of cells treated with hydrogen peroxide, we are able to extract regulatory events that are specific to ER stress and those that are shared with the oxidative stress response.

Using these data, we provide new insights into how the different regulatory levels affect gene expression in response to protein misfolding. We show that transcription regulation often occurs early in the stress response, while post-transcriptional regulation takes place both early and late during stress (*Figure 4*) - expanding on the canonical model of global translation shutdown followed by re-activation of protein synthesis. We observe both concordant and discordant regulation across different levels of regulation, that is processes that act in the same or opposite directions, respectively.

Several key regulators of the UPR are upregulated concordantly in their transcription and translation. The timing of their regulation supports the concept of an adaptive Unfolded Protein Response, where after surviving the acute phase of stress the cell adjusts via continual modulation of translation to recover protein synthesis and cope with the burdened ER (*Guan et al., 2017*; *Imrie and Sadler, 2012*; *Urra et al., 2013*). Translation of these genes increases early after tunicamycin treatment - during the acute phase - and is followed by a later increase in transcription, suggesting that the cell adapts to longer stress exposure through stepwise amplification of the expression response.

In comparison, we also find a statistically significant and unexpectedly large number of genes with discordance between their transcriptional and post-transcriptional regulation (*Figure 4*). These genes include DNA repair enzymes and mitochondrially localized proteins. We hypothesize on reasons behind these counterintuitive expression signatures below.

One set of genes is transcriptionally induced, but are unchanged or decrease in translation, and many of these genes function in DNA damage repair (*Figure 4*). We propose that these genes are regulated through transcriptional priming. Transcriptional priming has been well-established in plants where the transcriptome changes at the onset of stressful conditions and enables the fine-tuning of gene expression via translation later in the response (*Conrath et al., 2015*; *Hilker et al., 2016*). Indeed, translation decrease of DNA repair genes is consistent with suppressed double-strand break repair during ER stress (*Yamamori et al., 2013*) and likely achieved through the *PERK*-mediated pathways (*Oommen and Prise, 2013*). As the UPR can switch from pro-survival to pro-apoptosis during persisting stress (*Urra et al., 2013*), the cell might initially promote DNA repair through transcriptional priming, but dial down this response until it either reaches new homeostasis or the decision to initiate cell death is made.

## Translation regulation links energy metabolism, mitochondria, and ER stress

To the best of our knowledge, our results offer one of the first direct lines of support for translation, in addition to transcription, rerouting energy metabolism in response to ER stress, synthesizing recent disparate evidence for this connection (*Leibovitch and Topisirovic, 2018*; *Pascal and Boiteau, 2011*)(*Figure 5*). We find a surprising and significant number genes localized to mitochondria that are upregulated in translation in response to ER stress. Specifically, translation increases for genes involved in mitochondrial one-carbon metabolism (*SHMT2*, *MTHFD2*, *ALDH1L2*), but also in genes from serine biosynthesis (*PHGDH*, *PSAT1*, *PSPH*), which is upstream of one-carbon metabolism and localized to the cytosol. As serine biosynthesis diverts 3-phosphoglycerate from its use in glycolysis and the TCA cycle, the production of NADH via one-carbon metabolism may be used as an alternative to fuel Complex I and therefore drive oxidative phosphorylation (*Vazquez et al., 2011*). Instead of using glycolysis and the TCA cycle for NADH production, the cell employs serine biosynthesis and one-carbon metabolism to generate ATP during ER stress - and our results demonstrate that translation regulation supports this shift.

We suggest that this resource reallocation from glycolysis to one-carbon metabolism is linked to the increased need for the reducing agent glutathione (GSH) during ER stress (*Harding et al., 2003*). GSH is essential in maintaining redox balance by supporting the formation of disulfide bonds in the ER and preventing the accumulation of reactive oxygen species (*Figure 5C*)

(*Chakravarthi et al., 2006*). Hydrogen peroxide and other ROS can be produced directly during the enormous efforts of the cell to refold proteins during ER stress (*Zito et al., 2010*; *Guha et al., 2017*). Indeed, we find that protein disulfide isomerases, main players in protein folding, are translationally induced during ER stress. Their increased abundance will in turn heighten the demand for GSH in the ER to form disulfide bonds and in the cytosol to reduce the excess hydrogen peroxide.

Our data support several routes for the cell to accomplish this goal - mediated by translation. One route is through increased translation of *SHMT2* which synthesizes glycine that is required for the biosynthesis of GSH (*Amelio et al., 2014*; *Lu, 2013*). Another route is through translation induction of subunits of the ER translocon *SEC61* that directs the uptake of GSH into the ER (*Alder et al., 2005*; *Linxweiler et al., 2017*; *Ponsero et al., 2017*). Finally, we observe translation induction of *GPX1,* which utilizes GSH to reduce hydrogen peroxide (*Lubos et al., 2011*).

Collectively, the adjustment of translation might direct the flow of metabolism from glycolysis and the TCA cycle to serine biosynthesis and one-carbon metabolism during ER stress to not only support efficient synthesis of NADH for energy production by oxidative phosphorylation, but also to ensure GSH availability via glycine synthesis. Accordingly, we observe genes of the TCA cycle decrease in translation during ER stress, while genes of serine biosynthesis and one-carbon metabolism increase - providing unique evidence how translation links metabolism to the UPR.

The translation-mediated regulation of mitochondrially imported proteins also links to emerging evidence for another response pathway not considered in this work: the localization of mRNAs and, if appropriate, their translation near their proteins' target organelle. For example, during ER stress, mRNAs that encode proteins targeted to the ER are released from the ER membrane and decrease in their translation, while synthesis of cytosolic proteins is largely unchanged (*Reid et al., 2014*; *Guan et al., 2017*). Further, hundreds of mitochondrial proteins are synthesized by cytosolic ribosomes, but on mRNAs localized to the vicinity of the mitochondrial outer membrane (*Lesnik et al., 2015*). The relocalization of mRNAs in response to stress can therefore deliver another explanation for discordant regulation, for example transcription down-regulation but translation up-regulation of the mitochondrial genes, and provides an exciting area for future investigation.

## High-resolution data generate hypotheses on new regulators of the stress response and disease-relevant pathways

Our RNA-protein interaction maps for ribosomes and other *trans*-acting factors generate new hypotheses on mechanisms that underlie the regulation of expression signatures observed. As an example, we show that mRNA regions in the 3' UTR predicted to contain conserved secondary structures are preferentially occupied by proteins (*Figure 7*). While this finding provides promising evidence that these secondary structures can play a role in RNA-protein interactions, without further experimental validation via structural probing methods, for example selective 2'-hydroxyl acylation analyzed by primer extension (SHAPE)(*Wan et al., 2014*), we cannot be certain that these regions in fact fold in vivo.

Some of the RNA secondary structures are enriched for binding motifs of well-characterized RNA-binding proteins including Pumilio. However, Pumilio is known to bind single stranded RNA leaving it unclear how RNA structures may impact the binding of this protein. Pumilio is a well-characterized post-transcriptional regulator across different conditions thought to repress translation and localize to stress granules (*Kurisaki et al., 2009*; *Qiu et al., 2012*; *Weidmann et al., 2014*; *Morris et al., 2008*; *Namkoong et al., 2018*). While it is possible that the folding status of these 3' UTR regions may play a role in altering the accessibility of binding motifs described here, this remains speculation until further studies are performed.

We also observe stress-dependent RNA-protein interaction that coincides with induced translation, including for the chaperone *HSP90B1* (*GRP94*) that has multiple functions in the ER and stress response (*Eletto et al., 2010*). Our analysis identified significant translation upregulation for *HSP90B1* at eight hours after tunicamycin treatment, and this result coincides with increased protein binding to a conserved secondary structure in *HSP90B1*'s 3' UTR (*Figure 7*). Using a SHAPE-directed RNA folding algorithm (*Wan et al., 2014*; *Deigan et al., 2009*; *Lorenz et al., 2016*; *Zarringhalam et al., 2012*) to inform predictions of RNA structures, we observe that the region indeed has a high probability of forming a stem loop.

Similarly, we observe such translation increase correlating with changes in protein binding to 3' UTR structures for the chaperones *HSPA5* and *CALR* (*Figure 7—figure supplement 3*). As all three

examples are UPR regulators (*Vervliet et al., 2012*; *Yoshida, 2007*), we hypothesize on a new pathway affecting expression of these genes. Upon acute ER stress, the mRNAs for *HSP90B1*, *HSPA5*, and *CALR* are known to be released from the ER and translated in the cytosol (*Reid et al., 2014*). This re-localization is independent of signal sequences and its regulators are unknown (*Pyhtila et al., 2008*). It is tempting to speculate that the protein binding we observe at these conserved structures might represent the missing link that drives translocation and translation induction of these genes during stress. The relocalization of these induced illustrates this workflomRNAs is another example for additional pathways that can explain unconventional relationships observed between transcription and translation.

In sum, the results described here underscore the importance of integrating information from multiple levels of regulation to ascertain a comprehensive picture of the cellular response to stress. ER stress is associated with a wide range of human diseases, for example cancer, neurodegeneration, and liver disease (*Lin et al., 2008*). Indeed we find many genes involved in Alzheimer's, Parkinson's, and Huntington's disease to be induced in their translation (*Figure 3—figure supplement 7*). Other pathways we investigate in more detail, including those from serine biosynthesis and one-carbon metabolism, have strong links to cancer growth and survival (*Amelio et al., 2014*; *Maddocks et al., 2013*; *Mattaini et al., 2016*; *Yang and Vousden, 2016*). Finally, the transcriptional priming that we discuss has direct implications in various pathologies: mild ER stress exposure can enhance the cell's ability to respond to later, additional stress and protect against several severe disease phenotypes (*Inagi et al., 2008*; *Vacaru et al., 2014*; *Hara et al., 2011*; *Mendes et al., 2009*). The transcriptional induction of DNA repair genes soon after tunicamycin treatment might be one way for the cell to achieve such stress protection. As the UPR acts as an essential switch between cellular survival and death (*Schönthal, 2012*; *Cubillos-Ruiz et al., 2017*; *Scheper and Hoozemans, 2015*), our findings offer potential routes for developing new therapeutic strategies to promote survival among healthy cells that struggle to cope with such insults to the proteome, as well as drive apoptosis among tumorigenic cells known to hijack the UPR to promote their own growth. Taken together our results will support future analyses across a wide range of topics in the biomedical field.

# Materials and methods

## Key resources table

| Reagent type (species) or resource | Designation | Source or reference | Identifiers | Additional information |
|---|---|---|---|---|
| Cell line (H. sapiens) | CCL2 | ATCC Cat# CCL-2, RRID:CVCL_0030 | Authentication through Genetica Cell Line Testing | Authenticated by STR profiling |
| Antibody | Anti PERK | Cell Signaling technology | CST C33E10, RRID:AB_2095847 | (1:2000) |
| Antibody | anti-Phospho- PERK | Cell Signaling Technology | CST 3179S, RRID:AB_2095853 | (1:2000) |
| Antibody | anti-P-eIF2alpha | Cell Signaling Technology | CST 9721S, RRID:AB_330951 | (1:2000) |
| Antibody | Anti- β- Actin | Cell Signaling Technology | CST 4967S, RRID:AB_330288 | (1:5000) |
| Antibody | anti Oxidative Stress Markers | Abcam | Ab179843, RRID:AB_2716714 | (1:1000) |
| Other | CM-H2DCFDA | Thermo Fisher Scientific | C6827 | 10 µM |
| Sequence-based reagent | RPL19_ F 5'-ATGTATCACAGCCTGTACCTG-3'; RPL19_ R 5'-TTCTTGGTCTCTTCCTCCTTG-3' | PMID: 19137072 | | |
| Sequence-based reagent | sXBP1 forward 5' TGCTGAGTCCGCAGCAGGTG-3'; reverse 5'-GCTGGCAGGCTCTGGGGAAG-3') | PMID: 22038282 | | |

*Continued on next page*

*Continued*

| Reagent type (species) or resource | Designation | Source or reference | Identifiers | Additional information |
|---|---|---|---|---|
| Commercial assay or kit | Pierce Quantitative Colorimetric Peptide Assay | Thermo Fisher Scientific | 23275 | |
| Commercial assay or kit | SuperScript III First-Strand Synthesis System | Thermo Fisher Scientific | 18080051 | |
| Commercial assay or kit | KAPA SYBR FAST qPCR Master Mix (2X) Kit | Kapa Biosystem | KR0389 – v10.16 | |
| Commercial assay or kit | TMT 10plex | Sigma-Aldrich | 90110 | |
| Commercial assay or kit | Plasmotest | InvivoGen | rep-pt1 | |
| Commercial assay or kit | Universal Mycoplasma Detection Kit ATCC | ATCC | 30–1012K | |
| Chemical compound, drug | Hydrogen Peroxide | Sigma-Aldrich | 216763 | |
| Chemical compound, drug | Tunicamycin | Sigma-Aldrich | T7765 | |
| Chemical compound, drug | 2,2,2-Trifluoroethanol | Sigma-Aldrich | T8132 | |
| Chemical compound, drug | Iodoacetamide | Sigma-Aldrich | I6709 | |
| Software, algorithm | Maxquant (1.5.5.1) | PMID: 19029910 | RRID:SCR_014485 | |
| Software, algorithm | PECAplus | PMID: 24229407, 29263799 | https://github.com/PECAplus | |

## Cell culture and treatment

All samples for the total RNA, ribosome footprinting, protein occupancy profiling, and proteomics were derived from cells grown in parallel, arising from the same passage number. Cells were split just before the experiment and grown in parallel under identical conditions. Due to required protocols, we then prepared the samples for the total RNA-seq and ribosome profiling together, while the samples for the protein occupancy profiling and proteomics were processed separately. Biological replicates were collected independently on different days.

We grew Hela cells under standard condition,that is in DMEM (Sigma) with 10% fetal bovine serum (Atlanta biologicals) and 1X penicillin streptomycin solution (Corning cellgro) at 37°C and 5% CO2. At ~60% confluency, we treated the cells with 60 µM $H_2O_2$ or 0.5 µg/ml tunicamycin to induce oxidative and ER stress, respectively. We treated samples 8, 4, 1, and 0 hr prior to collection and therefore collected all samples at the same time with similar confluency. For protein occupancy profiling, we added 200 µM of 4-thiouridine at ten hours before the treatment to incorporate photoreactive ribonucleoside analog required for protein occupancy profiling. For ribosome footprinting, we added 0.1 mg/ml cycloheximide for 5 min at 37°C before the harvesting the cells.

We authenticated the cell line identity by STR profiling from Genetica DNA Laboratories. No mycoplasma contamination has been detected, as confirmed by mycoplasma contamination test using PlasmoTest kit (Invivogen) and ATCC universal mycoplasma detection kit (ATCC) separately.

## Sample collection for total RNA and ribosome footprinting

All steps were performed according the the TruSeq Ribo Profile (Mammalian) kit protocol. Confluent plates of HeLa cells were first aspirated of their growth media and washed with fresh media supplemented with 0.1 mg/ml cycloheximide. The media was then removed and the cells were washed with 10 ml chilled phosphate buffer saline (PBS) containing 0.1 mg/ml cycloheximide. Following removal of the PBS, 800 µl of Lysis Buffer was added and the cells were extensively scraped off plate and transferred to a pre-chilled eppendorf tube, recovering ~1 ml of lysate per sample. To insure complete lysis, we passed the lysate through a 25 gauge needle and further incubated on ice for ~10 min. The lysate was then clarified by centrifugation for 10 min at 20,000, 4°C and ~1 ml

supernatant was recovered. For each treatment (tunicamycin and hydrogen peroxide), we aliquoted 100 µl of the cell lysate for total RNA extraction and 200 ul for ribosome footprinting.

## Sample preparation

### Total RNA profiling and ribosome footprinting

We used the TruSeq Ribo Profile (Mammalian) kit for both total RNA profiling and mapping of ribosome protected fragments (RPF). Briefly, for the total RNA samples, 100 µl aliquots of lysate were supplemented with 10 µl of 10% SDS and purified using a Zymo RNA Clean and Concentrator-25 Kits. For the RPF samples, we treated the cell lysate with 5 units of Truseq Ribo Profile Nuclease for each A260 of lysate and incubated for 45 mins at room temperature while agitating gently. We stopped reactions by adding SUPERase-In RNase inhibitor and placed the samples on ice. The nuclease-treated lysate (100 µl) was purified using Illustra Microspin S-400 columns as per instructions followed by addition of 10 µl of 10% SDS to the flow-through. The remaining 100 µl of the nuclease digested RNA was stored in −80C for future use. Samples were purified using Zymo RNA Clean and Concentrator-25 kits and separated by denaturing 15% Urea-polyacrylamide gel electrophoresis (PAGE). We excised the desired size - corresponding to the ~28 – 30 nt range - using a dark field trans illuminator and purified according to the manufacturer's protocol.

For library preparation, all samples were RiboZero treated and the total RNA samples were heat-fragmented at 94°C for 25 min according to the manufacturer's protocol. Both the fragmented total RNA and RPF samples were then end-repaired and 3' adapter ligated. Following adapter removal, the samples were reverse transcribed and the resulting cDNA was PAGE purified and circularized. We used one quarter of the circularized cDNA as a template for PCR amplification using Phusion (NEB) in a 50 µl volume using specific oligos (Truseq Riboprofile Forward PCR primer and Index PCR primer) and purified the samples with Agencount AMPure beads (Beckman Coulter). We purified the amplified libraries by using 8% Native PAGE and verified the final library size using the High Sensitivity DNA assay on the Agilent Bioanalyzer. We quantified samples using Qubit Fluorometric quantitation and sequenced them on a HiSeq 2500.

### Protein occupancy profiling

After sample collection, cells were crosslinked with 365 nm UV light (0.2 J/cm2) on ice using a Stratalinker 2400 (Stratagene). We scraped crosslinked cells off the plates with a rubber policeman, collected by centrifugation, washed with ice-cold PBS once and flash-froze the samples in liquid nitrogen for long-term storage.

Protein occupancy profiling was carried out as described previously (*Munschauer, 2015*). Briefly, we resuspended cell pellets in lysis/binding buffer (100 mM Tris-HCl pH 7.5 at 25°C, 500 mM LiCl, 10 mM EDTA pH 8.0 at 25°C, 1% LiDS, 5 mM DTT, Complete Mini EDTA-free protease inhibitor (Roche), incubated them at room temperature for 15 min for lysis and passed cells through a 21 gauge needle 10 times for shearing of genomic DNA. Lysates were incubated with oligo(dT) Dynabeads (Ambion) for 1 hr at room temperature on a rotating wheel. Following incubation, the beads were concentrated on a magnetic rack and the supernatant was stored on ice for further rounds poly (A)$^+$-RNA depletion. We washed beads 3 times in lysis/binding buffer and 3 times in NP40 washing buffer (50 mM Tris-HCl pH 7.5 at 25°C, 140 mM LiCl, 2 mM EDTA pH 8.0 at 25°C, 0.5% NP40, 0.5 mM DTT) and crosslinked poly(A)$^+$-RNA-protein complexes were eluted in low-salt elution buffer (10 mM Tris-HCl at at 25°C) by incubation at 80°C for 2 min.

The stored supernatants were re-incubated with beads for two additional rounds of poly(A)$^+$-RNA depletion following the described procedure. Eluates from different rounds of poly(A)$^+$-RNA depletion are combined, incubated with RNase I for 10 min at 37°C and precipitated with 4 volumes of ammonium sulfate. The resuspended precipitate was separated by SDS-PAGE and transferred onto a nitrocellulose membrane. RNA-protein complexes were on-membrane incubated with Proteinase K for 30 min at 55°C to release protein-protected RNA fragments. RNA was recovered with phenol-chloroform extraction and subjected to small RNA library cloning procedure (*Hafner et al., 2012*). In short, RNA fragments generated by RNase I digestion were dephosphorylated with Calf intestinal alkaline phosphatase (CIP), radiolabeled at the 5' end with [γ-$^{32}$P]-ATP in a T4 Polynucleotide kinase (PNK) reaction followed by ligation of a pre-adenylated 3' adapter and a 5' adapter and reverse transcription to generate the cDNA library. In order to identify the RNA population with the desired

fragment size, radiolabeled RNA size markers (24 and 50 nt) were used as ligation controls. The cDNA libraries were processed and subjected to sequencing on HiSeq 2500 following standard protocol (*Hafner et al., 2012*).

## Proteomics analysis by mass spectrometry

For proteomics analysis, we resuspended cell pellets for each sample in 50 µl ice-cold PBS containing 1:100 protease inhibitor cocktail. The cells were then sonicated with probe sonicator 2 × 30 s with amplitude 5. The samples were returned to ice for 30 s between sonication intervals. After sonication, we mixed the samples with 50 µl trifluororethanol, and the mixtures were kept at 60°C for 1 hr. Then the samples were reduced in 15 mM DTT at 55°C for 45 min, and alkylated in 55 mM iodoacetamide in the dark at room temperature for 30 min. Finally, we used 50 mM Tris (pH = 8) to adjust the sample volume to 1 ml, and 1 ug mass spectrometry grade trypsin (Sigma Aldrich) was added to digest the proteins into peptides at 37°C overnight.

We measured peptide concentrations with the Pierce Quantitative Fluorometric Peptide Assay (ThermoFisher) kit a. Tandem mass tag (TMT) 10-plex reagents (Thermo Scientific) were dissolved in anhydrous acetonitrile (0.8 ug/40 µl) according to manufacturer's instruction. We labeled 30 ug/100 µl peptide per sample labelled with 41 µl of the TMT 10-plex label reagent at final acetonitrile concentration of 30% (v/v). Following incubation at room temperature for 1 hr, we quenched the reactions with 8 µl of 5% hydroxylamine for 15 min. All samples were combined in a new microcentrifuge tubes at equal amounts and reduced to remove acetonitrile using an Eppendorf Concentrator Vacufuge Plus.

TMT-labelled tryptic peptides were subjected to high-pH reversed-phase high performance liquid chromatography fractionation using an Agilent 1200 Infinity Series with a phenomenex Kinetex 5 u EVO C18 100A column (100 mm x 2.1 mm, 5 mm particle size). Mobile phase A was 20 mM ammonium formate, and B was 90% acetonitrile and 10% 20 mM ammonium formate. Both buffers were adjusted to pH 10. Peptides were resolved using a linear 120 min 0 – 40% acetonitrile gradient at a 100 ul/min flow rate. Eluting peptides were collected in 2 min fractions. We combined about 70 fractions covering the peptide-rich region to obtain 40 samples for analysis. To preserve orthogonality, we combined fractions across the gradient, that is each of the concatenated samples comprising fractions which were 40 fractions apart. Re-combined fractions were reduced using an Eppendorf Concentrator Vacufuge Plus, desalted with C18 stage-tip, and suspended in 95% mass spectrometry grade water, 5% acetonitrile, and 0.1% formic acid for subsequent low pH chromatography and tandem mass spectrometry analysis.

For the first replicate, we used an EASY-nLC 1200 coupled on-line to a Fusion Lumos mass spectrometer (both Thermo Fisher Scientific). Buffer A (0.1% FA in water) and buffer B (0.1% FA in 80% ACN) were used as mobile phases for gradient separation. A 75 µm x 15 cm chromatography column (ReproSil-Pur C18-AQ, 3 µm, Dr. Maisch GmbH, German) was packed in-house for peptides separation. Peptides were separated with a gradient of 5 – 40% buffer B over 110 min, 40 – 100% B over 10 min at a flow rate of 300 nL/min. Full MS scans were acquired in the Orbitrap mass analyzer over a range of 300 – 1500 m/z at a resolution of 120,000 at m/z 200. The top 15 most abundant precursors were selected in data-dependent mode with an isolation window of 0.7 Thomsons and fragmented by higher-energy collisional dissociation with normalized collision energy of 40. MS/MS scans were acquired in the Orbitrap mass analyzer at a resolution of 30,000. The automatic gain control target value was 1e6 for full scans and 5e4 for MS/MS scans respectively, and the maximum ion injection time is 60 ms for both.

For the second replicate, we used an EASY-nLC 1000 coupled on-line to a Q Exactive spectrometer (both Thermo Fisher Scientific). Buffer A (0.1% FA in water) and buffer B (80% acetonitrile, 0.5% acetic acid) were used as mobile phases for gradient separation. An EASY Spray 50 cm x 75 µm ID PepMap C18 analytical HPLC column with 2 µm bead size was used for peptide separation. We used a 110 min linear gradient from 5% to 23% solvent B (80% acetonitrile, 0.5% acetic acid), followed by 20 min from 23% to 56% solvent B, and 10 min from 56% to 100% solvent B. Solvent B was held at 100% for another 10 min. Full MS scans were acquired with a resolution of 70,000, an AGC target of 1e6, with a maximum ion time of 120 ms, and scan range of 400 to 1500 m/z. Following each full MS scan, data-dependent high resolution HCD MS/MS spectra were acquired with a resolution of

35,000, AGC target of 1e5, maximum ion time of 250 ms, one microscan, 1.5 m/z isolation window, fixed first mass of 115 m/z, and NCE of 30.

## Measurement of reactive oxygen species

We used CM-H2DCFDA (Thermo Fisher Scientific) following protocols in (*Poursaitidis et al., 2017*). Briefly, after tunicamycin or hydrogen peroxide treatment, we washed the cells with PBS and loaded with CM-H2DCFDA (10 µM) in Dulbecco's PBS for 90 mins, trypsinized with 0.25% Trypsin-EDTA, resuspended in PBS with 10% fetal bovine serum, and analyzed using an BD Acuri (Becton Dickinson). Dyes were excited using a blue 488 nm laser, and emission was recorded on FL1 (514/20) for a minimum of 5,000 cells per sample. Small cellular debris was excluded by gating on a forward scatter plot.

## Validation experiments

### qRT-PCR

For qPCR, we followed manufacturer's instructions unless noted otherwise. We isolated total RNA from Hela cells treated with tunicamycin using Trizol extraction (Thermo Fisher Scientific) and purified RNA with the RNAesay mini Kit (Qiagen). We then synthesized cDNA using the Super ScriptIII First Strand cDNA synthesis kit (Invitrogen, Life technologies). We estimated *sXbp1* quantities by SYBR Green quantitative real-time PCR using Kapa Universal SYBR Green Supermix (Kapa biosystem) in a Roche Light Cycler 480 (Roche). All reactions were performed in triplicate. The expression of the spliced XBP1 was assessed by real time PCR (RT-PCR) in a Roche Light Cycler 480 (Roche) with KAPA universal SYBR green master mix. Relative gene expression was quantified using the ΔCT method with respective primers (*sXBP1* forward 5′-TGCTGAGTCCGCAGCAGGTG-3′; reverse 5′-GCTGGCAGGCTCTGGGGAAG-3′) and normalized to *RPL19* (forward 5′-ATGTATCACAGCCTGTACCTG-3′; reverse 5′-TTCTTGGTCTCTTCCTCCTTG-3′). We used ΔΔCT method to determine the fold changes in the expression of sXbp1 (*Livak and Schmittgen, 2001*). Briefly, the threshold cycle (Ct) was determined and relative gene expression was calculated as follows: fold change = 2-Δ(ΔCt), where ΔCt (cycle difference)=Ct(target gene)-Ct(control gene) and Δ(ΔCt)=Ct(treated condition)-Ct (control condition).

### Western blotting

We boiled all samples collected at different time points after indicated treatment in sample buffer (Bio-Rad) supplemented with B-mercaptoethanol as per manufacturer's instruction. We used equal amount of protein (25 µg) from three independent grown cultures and treatments for blotting. The membrane was blocked using 5% BSA and incubated with respective antibodies (rabbit PERK (1:2000, CST C33E10), rabbit P-PERK (1:2000, CST 3179), rabbit P- eIF2alpha (1:2000, CST 9721S), rabbit anti Oxidative Stress Markers (Abcam ab179843). β- Actin (1:5000, CST 4967S) served as a loading control. We captured signal intensities of the bands in the Western blot with Kwikquant Imager (Kindle Biosciences, USA).

## Quantification and statistical analyses

### RNA, ribosomal and (non-ribosomal) protein footprints

We used an in-house R pipeline to process the sequencing data from total RNA, ribosome footprinting and protein occupancy profiling. The fastq files of individual samples were processed with FASTX-Toolkit v0.0.14 to remove adapters and trim reads based on a minimum quality score of 20, as well as discard reads with a trimmed length shorter than 20 nucleotides. Reads mapping to ribosomal DNA were removed using Bowtie2 (*Langmead and Salzberg, 2012*). Remaining processed reads were aligned to the human reference genome (hg19) with TopHat v2.1.1 (*Trapnell et al., 2012*) using the gencode v19 GTF reference transcriptome (*Harrow et al., 2012*). Aligned bam files were filtered based on unique mapping and read length: RNA and protein footprints > 25 nt, ribosome footprints 28–30 nt. Total counts per gene were calculated using HTseq (*Planet et al., 2012*). For expression measurements comprising the core dataset (*Figure 2*), RNA reads aligning to exons were counted as a measure of processed mRNA abundance, ribosomal footprint reads mapping to the coding regions of transcripts (CDS) were counted as a measure of translating ribosomes, and protein footprint reads mapping to the untranslated regions (UTR) were counted as a measure of

protein bound UTR. Genes were filtered based on a minimum count of 10 across all samples and counts were normalized using the median ratio method implemented in DESeq2 (*Anders and Huber, 2010*; *Love et al., 2014*). Surrogate variables were estimated and removed via linear modeling (SVA) to remove batch effects (*Leek and Storey, 2007*). The data includes a total of ~14,000 genes and is described in *Supplementary file 1–3*.

## Protein

We used MaxQuant Software version 1.5.5.1 with its integrated search engine Andromeda to analyse our raw files acquired from the mass spectrometer. Data was searched against the human sequence file (Homo_sapiens.GRCh37.75.pep.all.fa) sequence file downloaded from the ENSEMBL database (*Cox et al., 2011*; *Cox and Mann, 2008*; *Tyanova et al., 2016*). All sample fractions of two individual sets were grouped by setting up experimental design parameters in Maxquant. The mass tolerance of MS/MS spectra were set to 20 ppm with a posterior global FDR of 1% based on the reverse sequence of the human FASTA file. In addition, MS/MS data were searched by Andromeda for potential common mass spectrometry contaminants. Trypsin/P specificity was used to perform database searches, allowing two missed cleavages. Carbamidomethylation of cysteine residues and 10-plex TMT modifications on Lys and N-terminal amines were considered as a fixed modification, while oxidation of methionines and N-terminal acetylation were considered as variable modifications. TMT quantification was performed at MS2 level with default mass tolerance and other parameters. We then used the reporter ion intensities as estimates for protein abundance. A total of 10,399 protein groups were identified, including 0.01% reverse sequences and contaminants. Protein groups with no measurement among either replicate were then removed, as well as those identified by only one peptide in either replicate. After filtering, for each protein the geometric mean was calculated across all samples within one stress and the intensities were divided by this mean. The median of these ratios over all proteins was used as a size factor to account for differences in global sequence coverage between samples, similar to library size normalization for RNA sequencing and footprinting experiments. SVA was also applied to remove batch effects as described above (*Leek and Storey, 2007*). The data describes a total of N = 7255 protein groups and is presented in *Supplementary file 4*.

## The core dataset

To derive the core dataset of 7,011 genes (*Figure 3—figure supplement 5*), we first mapped all data to common ENSG identifiers. If several genes or isoforms mapped to the same data, we used the major/most abundant isoform as the group's identifier. The data was then filtered for completeness across all replicates and resulted in the 7,011 genes presented here as the core dataset. For each time series experiment, we calculated the log base 10 ratios of the measurement at time x compared to the measurement at time 0. We then normalized across the entire time course (but for each dataset separately) by subtracting the average value from each entry and dividing by the standard deviation.

We visualized and analyzed the data matrix in PERSEUS version 1.5.5.1 (*Tyanova and Cox, 2018*), using hierarchical clustering, the 'Correlation' and 'Complete' options, marking values in blue-white-red scale (*Figure 3—figure supplement 1*). The data discussed in the main text focuses on two of the replicates and the first six eigenvectors of the subsequent PCA of the data. Using PERSEUS, we clustered the core dataset into 20 clusters with highest coherence in the similarity measure. The functional analyses were generated through the use of IPA (QIAGEN Inc., https://www.qiagenbio- informatics.com/products/ingenuity-pathway-analysis), with an adjusted p-value cutoff of 0.0001.

## Extracting significant regulatory events via Protein Expression Control Analysis (PECA)

To extract significant regulatory events, we adapted the statistical tool PECA that we developed and expanded recently (*Teo et al., 2018*; *Teo et al., 2014*). For PECA, we used the abundance data after SVA-based removal of surrogate variables, but prior to other transformations and normalization described above. The data was transformed by taking the natural logarithm and centered by subtracting the row median. We then smoothed the data using the Gaussian Process tool in PECAplus

with standard settings and replicate information. We then performed the PECA analysis using standard settings as described in reference (*Teo et al., 2018*; *Teo et al., 2014*) for absolute, label-free data. In brief, PECA extracts significant regulatory events for each gene and each time point by examining the overall noise in the data and specifically the *change* in the synthesis and degradation of the respective molecule since and until the time points before and after the time point in question. For the footprinting data, synthesis and degradation are replaced by association and dissociation of the respective molecules.

The paired concentration data for the four different levels was derived as follows: i) for transcription and RNA degradation (TRXP; RNA-DEG) we paired the RNA abundance data with DNA abundance set to a constant; ii) for translation (TRL) we paired the ribosome footprinting with the RNA abundance data; iii) for translation and RNA degradation/localization/processing (TRL; RNA-DEG) we paired the protein footprinting with the RNA abundance data; and iv) for translation and protein degradation (TRL; PROT-DEG) we paired the protein and the RNA abundance data.

PECA reports for each gene a putative change point score. Given the overall score distribution it also calculates a false discovery rate (FDR) for each data point. We reported a significant regulatory event for a gene if we observed a change point with a score corresponding to an FDR < 0.2 in both replicates, in the same direction (up or down), but regardless of the time point at which the event occurred. Stricter cutoffs resulted in very similar results (*Figure 3—figure supplement 2*). PECA results are provided in *Supplementary file 1–5*.

## Alternative splicing analysis

We generated Sashimi plots (*Katz et al., 2015*) for the 20 cytosolic aminoacyl-tRNA synthetases on the Integrative Genomics Viewer (IGV) from RNA-seq data for all four time points and searched manually for alternative splicing events. These events were marked by reads that spanned exon-exon junctions and therefore unambiguously denoted inclusion or exclusion of specific exons. Single read events were excluded. To reduce the number of false positive splicing events, we only counted junction reads that started at or ended on at least one known exon boundary as determined by the RefSeq variants on IGV. Exons that were not annotated in these transcript variants were only included in the analysis if they comprised junction reads on both exon boundaries. We recorded the events as a major (V1) and minor (V2) splicing event (*Supplementary file 6*). We then examined the data for putative stress-dependent changes in production of these variants, requiring consistency across the three replicates.

## Analysis of proteins binding to RNA secondary structures

We obtained a list of conserved RNA secondary structures from reference (*Parker et al., 2011*). Coverage of structures was calculated using htseq to count reads mapping within ±200 nucleotides of the midpoint. Structures were considered to be transcribed if they had an average coverage of 10 reads among untreated samples. Using R, we mapped these structures to the processed RNA, ribosome and protein footprinting data and evaluated the accumulation of reads around the structure (*Supplementary file 7*). We extracted motifs in the RNA sequence in and surrounding the secondary structure with the MEME package using standard settings (*Bailey et al., 2006*).

### Data availability

The data discussed in this publication have been deposited in NCBI's Gene Expression Omnibus (*Barrett et al., 2013*; *Edgar et al., 2002*) and are accessible through GEO Series accession number GSE113171. The mass spectrometry data including the MaxQuant output files have been deposited to the ProteomeXchange Consortium via the PRIDE (*Vizcaíno et al., 2016*) partner repository with the dataset identifier PXD008575.

## Acknowledgements

The work was supported by the NIH/NIGMS grants 1R01GM113237-01 and 1R35GM127089-01 (to CV). We thank Kirsten Sadler-Edepli for providing the list of known UPR genes.

## Additional information

### Competing interests
Lindsay Freeberg, Scott Kuersten: affiliated with Illumina Inc. No other competing interests to declare. The other authors declare that no competing interests exist.

### Funding

| Funder | Grant reference number | Author |
| --- | --- | --- |
| National Institutes of Health | 1R01GM113237-01 | Justin Rendleman<br>Zhe Cheng<br>Shuvadeep Maity<br>Guoshou Teo<br>Christine Vogel |
| National Institutes of Health | 1R35GM127089-01 | Justin Rendleman<br>Shuvadeep Maity<br>Christine Vogel |

The funders had no role in study design, data collection and interpretation, or the decision to submit the work for publication.

### Author contributions
Justin Rendleman, Conceptualization, Resources, Data curation, Software, Formal analysis, Validation, Investigation, Visualization, Methodology, Writing—original draft, Writing—review and editing; Zhe Cheng, Data curation, Validation, Methodology, Writing—review and editing; Shuvadeep Maity, Conceptualization, Data curation, Formal analysis, Validation, Investigation, Visualization, Methodology, Writing—original draft, Writing—review and editing; Nicolai Kastelic, Mathias Munschauer, Data curation, Formal analysis, Methodology, Writing—review and editing; Kristina Allgoewer, Data curation, Formal analysis, Investigation, Writing—original draft, Writing—review and editing; Guoshou Teo, Software, Methodology, Writing—review and editing; Yun Bin Matteo Zhang, Software, Formal analysis; Amy Lei, Data curation, Formal analysis, Visualization; Brian Parker, Data curation, Formal analysis, Visualization, Methodology; Markus Landthaler, Resources, Supervision, Funding acquisition, Writing—review and editing; Lindsay Freeberg, Scott Kuersten, Resources, Data curation, Formal analysis, Writing—review and editing; Hyungwon Choi, Conceptualization, Data curation, Software, Formal analysis, Funding acquisition, Investigation, Methodology, Writing—review and editing; Christine Vogel, Conceptualization, Resources, Data curation, Software, Formal analysis, Supervision, Funding acquisition, Validation, Investigation, Visualization, Methodology, Writing—original draft, Project administration, Writing—review and editing

### Author ORCIDs
Justin Rendleman http://orcid.org/0000-0001-8152-7127
Shuvadeep Maity http://orcid.org/0000-0002-6031-4744
Hyungwon Choi http://orcid.org/0000-0002-6687-3088
Christine Vogel http://orcid.org/0000-0002-2856-3118

### Decision letter and Author response
Decision letter https://doi.org/10.7554/eLife.39054.046
Author response https://doi.org/10.7554/eLife.39054.047

## Additional files

### Supplementary files
• Supplementary file 1. Input and output values for PECA analysis of RNA expression data. Content: Worksheet 1: Description: Notes on columns in other worksheets; Worksheet 2: Total_RNA_Tunicamycin_Rep1: log2 transformed normalized count values for tunicamycin treated replicate 1; Worksheet 3: Total_RNA_Tunicamycin_Rep2: log2 transformed normalized count values for tunicamycin

treated replicate 2; Worksheet 4: Total_RNA_Tunicamycin_Rep3: log2 transformed normalized count values for tunicamycin treated replicate 3; Worksheet 5: Total_RNA_H2O2_Rep1: log2 transformed normalized count values for hydrogen peroxide treated replicate 1; Worksheet 6: Total_RNA_-H2O2_Rep2: log2 transformed normalized count values for hydrogen peroxide treated replicate 2; Worksheet 7: Total_RNA_H2O2_Rep3: log2 transformed normalized count values for hydrogen peroxide treated replicate 3

DOI: https://doi.org/10.7554/eLife.39054.031

• Supplementary file 2. Input and output values for PECA analysis of ribosome footprinting data. Content: Worksheet 1: Description: Notes on columns in other worksheets; Worksheet 2: RPF_Tunicamycin_Rep1: log2 transformed normalized count values for tunicamycin treated replicate 1; Worksheet 3: RPF_Tunicamycin_Rep2: log2 transformed normalized count values for tunicamycin treated replicate 2; Worksheet 4: RPF_Tunicamycin_Rep3: log2 transformed normalized count values for tunicamycin treated replicate 3; Worksheet 5: RPF_H2O2_Rep1: log2 transformed normalized count values for hydrogen peroxide treated replicate 1; Worksheet 6: RPF_H2O2_Rep2: log2 transformed normalized count values for hydrogen peroxide treated replicate 2; Worksheet 7: RPF_H2O2_Rep3: log2 transformed normalized count values for hydrogen peroxide treated replicate 3

DOI: https://doi.org/10.7554/eLife.39054.032

• Supplementary file 3. Input and output values for PECA analysis of protein occupancy profiling data. Content: Worksheet 1: Description: Notes on columns in other worksheets; Worksheet 2: POP_Tunicamycin_Rep1: log2 transformed normalized count values for tunicamycin treated replicate 1; Worksheet 3: POP_Tunicamycin_Rep2: log2 transformed normalized count values for tunicamycin treated replicate 2; Worksheet 4: POP_Tunicamycin_Rep3: log2 transformed normalized count values for tunicamycin treated replicate 3; Worksheet 5: POP_H2O2_Rep1: log2 transformed normalized count values for hydrogen peroxide treated replicate 1; Worksheet 6: POP_H2O2_Rep2: log2 transformed normalized count values for hydrogen peroxide treated replicate 2; Worksheet 7: POP_H2O2_Rep3: log2 transformed normalized count values for hydrogen peroxide treated replicate 3

DOI: https://doi.org/10.7554/eLife.39054.033

• Supplementary file 4. Input and output values for PECA analysis of protein expression data. Content: Worksheet 1: Description: Notes on columns in other worksheets; Worksheet 2: Protein_Tunicamycin_Rep1: log2 transformed normalized intensity values for tunicamycin treated replicate 1; Worksheet 3: Protein_Tunicamycin_Rep3: log2 transformed normalized intensity values for tunicamycin treated replicate 3; Worksheet 4: Protein_H2O2_Rep1: log2 transformed normalized intensity values for hydrogen peroxide treated replicate 1; Worksheet 5: Protein_H2O2_Rep3: log2 transformed normalized intensity values for hydrogen peroxide treated replicate 3; Worksheet 6: All_identified_Protein_Groups: Maxquant output with detailed information of all the protein groups identified; Worksheet 7: Identified_Protein_Groups_w_rev: Maxquant output with detailed information of all the protein groups identified along with reverse peptide information

DOI: https://doi.org/10.7554/eLife.39054.034

• Supplementary file 5. Integrated, post-processed core data set (7,011 genes). Content: Worksheet 1: Description: Notes on columns in other worksheets; Worksheet 2: Processed_core_data: core data set (Z-score); Worksheet 3: Cluster_Analysis_Canonical_Path: list of enriched pathways with P-values corresponding

DOI: https://doi.org/10.7554/eLife.39054.035

• Supplementary file 6. Extended data for aminoacyl-tRNA synthetase (AAtRS) analysis. Content: Worksheet 1: Cytosolic_AAtRS: Raw count values for cytosolic AAtRS related to *Figure 6*; Worksheet 2: Abbreviations: Details of the abbreviations used in Worksheet 1

DOI: https://doi.org/10.7554/eLife.39054.036

• Supplementary file 7. Extended RNA secondary structures data used in *Figure 7*. Content: Worksheet 1: Description: Notes on columns in other worksheets; Worksheet 2: 5'UTR_Structure_RibosomeBound: Includes values for mid point position of predicted structure and Base mean (mean normalized count values) of ribosome footprints; Worksheet 3: 3'UTR_Structure_ProteinBound: Includes values for mid point position of predicted structure and Base mean (mean normalized count values) of protein footprints on RNA

DOI: https://doi.org/10.7554/eLife.39054.037

• Transparent reporting form
DOI: https://doi.org/10.7554/eLife.39054.038

## Data availability

The data discussed in this publication have been deposited in NCBI's Gene Expression Omnibus (Barrett et al., 2013; Edgar et al., 2002) and are accessible through GEO Series accession number GSE113171. The mass spectrometry data including the MaxQuant output files have been deposited to the ProteomeXchange Consortium via the PRIDE (Vizcaíno et al., 2016) partner repository with the dataset identifier PXD008575.

The following dataset was generated:

| Author(s) | Year | Dataset title | Dataset URL | Database, license, and accessibility information |
|---|---|---|---|---|
| Justin Rendleman, Zhe Cheng, Shuva-deep Maity, Nicolai Kastelic, Mathias Munschauer, Kristina Allgoewer, Guoshou Teo, Yun Bin Matteo Zhang, Amy Lei, Brian Parker, Markus Landthaler, Lindsay Freeberg, Scott Kuersten, Hyungwon Choi, Christine Vogel | 2018 | Data from: New insights into the cellular temporal response to proteostatic stress | http://www.ncbi.nlm.nih.gov/geo/query/acc.cgi?acc=GSE113171 | Publicly available at the NCBI Gene Expression Omnibus (accession no. GSE113171) |

The following previously published datasets were used:

| Author(s) | Year | Dataset title | Dataset URL | Database, license, and accessibility information |
|---|---|---|---|---|
| Wan Y, Qu K, Zhang QC, Flynn RA, Manor O, Ouyang Z, Zhang J, Snyder MP, Segal E, Chang HY | 2013 | Landscape and variation of RNA secondary structure across the human transcriptome | https://www.ncbi.nlm.nih.gov/geo/query/acc.cgi?acc=GSE50676 | Publicly available at the NCBI Gene Expression Omnibus (accession no: GSE50676). |
| Berkowitz ND, Silverman IM, Childress DM, Kazan H, Wang L, Gregor BD | 2016 | A comprehensive database of high-throughput sequencing-based RNA secondary structure probing data (Structure Surfer) | https://www.ncbi.nlm.nih.gov/geo/query/acc.cgi?acc=GSE72681 | Publicly available at the NCBI Gene Expression Omnibus (accession no: GSE72681). |

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
