## [Decision Letter]

Thank you for submitting your article "Systematic characterization of the adaptation to proteostatic challenges" for consideration by *eLife*. Your article has been reviewed by three peer reviewers, including Juan Valcárcel as the Reviewing Editor and Reviewer #1, and the evaluation has been overseen by Naama Barkai as the Senior Editor.

The reviewers have discussed the reviews with one another and the Reviewing Editor has drafted this decision to help you prepare a revised submission.

The manuscript by Justin Rendleman and coauthors is an integrated quantitative study that systematically captures transcriptional and post-transcriptional changes induced in HeLa cells upon exposure to two cellular stresses, tunicamycin (which induces ER stress) and H_2_O_2_ (oxidative stress). The data represent a technical tour-de-force in the global characterization of the regulation of the response of human cells to proteostatic stress, and the authors provide good examples of the validity and usefulness of the datasets. The findings bring new insights in our understanding of the cellular response to stress and are expected to be of high value in a broad range of research fields that study physiological or pathological responses to stress conditions.

Major comments:

1) The way in which the interplay between ER stress and oxidative stress is discussed in different points of the paper is problematic. At some points, such as when referring to Figures 1B and 2, oxidative stress is presented as having very little effect under the experimental conditions used by the authors ("In comparison, oxidative stress affects translation very little in our experiment"). In contrast, somewhere else in the paper the opposite seems to be said, explicitly such as in the third paragraph of the Discussion ("All levels of regulation include genes that respond similarly in both stresses") or implicitly in many parts of the paper. This is confusing and should be clarified.

2) The wide diversity of ways in which different results are displayed can be distracting. For instance, Figure 1A introduces the four data types and their representation in one way, but different representation methods are used in the results shown in Figure 1B on the one hand, Figure 1C-E on the other, and Figure 1F on yet another. The same can be said of the rest of the figures. Certainly, any paper that addresses a global analysis of cellular responses can be expected to have large amounts of data, but this highlights even more the importance of being clear in how this data is presented, in order to make the final take-home message of the paper as evident to the reader as possible.

3) Data in Figure 5 are based on predictions of RNA secondary structures, which are largely based on RNA stability estimates of relatively long fragments and phylogenetic conservation of base pairing potential; these predictions are necessarily imprecise and unlikely to reflect the actual structure of RNA within complex RNP particles at 5' and 3' UTRs. While detection of prominent structures like iron response elements is reassuring, how predictable other structural elements are in the absence of in vivo RNA structure data (e.g. using SHAPE) should be discussed properly and, if possible, stratified in terms of predicted stability of secondary elements/conservation. It will be also relevant to discuss the predicted local structure at the site of interaction with proteins displaying enriched sequence motifs: the proteins discussed in the text bind to single stranded RNA: is the local structure at these sites predicted to be open within the secondary structure of these elements? By the way, Sex-Lethal is a *Drosophila* protein without a homolog in human cells, although a number of poly-U binding proteins (including ELAV-like, HuD and related factors) share similar sequence motifs. Therefore the presence of a SXL motif should be explained in these terms in the text and in Figure 5B.

---

## [Author Response]

Major comments:1) The way in which the interplay between ER stress and oxidative stress is discussed in different points of the paper is problematic. At some points, such as when referring to Figures 1B and 2, oxidative stress is presented as having very little effect under the experimental conditions used by the authors ("In comparison, oxidative stress affects translation very little in our experiment"). In contrast, somewhere else in the paper the opposite seems to be said, explicitly such as in the third paragraph of the Discussion ("All levels of regulation include genes that respond similarly in both stresses") or implicitly in many parts of the paper. This is confusing and should be clarified.

We restructured the manuscript to clarify our focus on the ER stress response and the connections to oxidative stress (Introduction, seventh paragraph, and Results subsections “Multi-layered data types identify new regulatory signatures during stress” and “The shared stress response involves few genes but highly similar patterns”).

We now discuss the results for the response to hydrogen peroxide in a separate section and explain that many changes are qualitatively similar but much less pronounced than in ER stress (subsection “The shared stress response involves few genes but highly similar patterns”).

The revised manuscript also contains a schematic explaining the relationship between ER stress, oxidative stress, and cancer cells (new Figure 1A) and graphs with the measurements of the ROS levels as discussed below (Figure 2A).

2) The wide diversity of ways in which different results are displayed can be distracting. For instance, Figure 1A introduces the four data types and their representation in one way, but different representation methods are used in the results shown in Figure 1B on the one hand, Figure 1C-E on the other, and Figure 1F on yet another. The same can be said of the rest of the figures. Certainly, any paper that addresses a global analysis of cellular responses can be expected to have large amounts of data, but this highlights even more the importance of being clear in how this data is presented, in order to make the final take-home message of the paper as evident to the reader as possible.

We reworked the figures to clarify the display of the data and results. Wherever possible, we now consistently show heatmaps and moved the original figures to the supporting materials (Figures 3-7).

The new overview figure (Figure 1B) explains the workflow with heatmaps and we moved the original Figure 1A – which explains PECA – to the supporting material (Figure 1—figure supplement 1). This change aims at clarifying that PECA was used to extract significant regulatory events (for each gene and time point), but algorithm of the method has been published elsewhere and should not be the focus of this manuscript.

All validation experiments are now in a separate figure (new Figure 2). While we moved the genomic traces of ATF4 and GADD34 to the supporting material (Figure 2—figure supplement 4), the other validation experiments show a variety of plots which we consider important orthogonal experimental approaches to confirm that we correctly induced ER stress and its relationship to the oxidative stress experiment.

We adjusted the text according to these changes to the figures (Results subsections “Multi-layered data types identify new regulatory signatures during stress” and “The shared stress response involves few genes but highly similar patterns”). Other measures to increase clarity of the manuscript include editing of the Abstract and discussing the oxidative stress response in a separate section, addressing another comment below (subsection “The shared stress response involves few genes but highly similar patterns”).

3) Data in Figure 5 are based on predictions of RNA secondary structures, which are largely based on RNA stability estimates of relatively long fragments and phylogenetic conservation of base pairing potential; these predictions are necessarily imprecise and unlikely to reflect the actual structure of RNA within complex RNP particles at 5' and 3' UTRs. While detection of prominent structures like iron response elements is reassuring, how predictable other structural elements are in the absence of in vivo RNA structure data (e.g. using SHAPE) should be discussed properly and, if possible, stratified in terms of predicted stability of secondary elements/conservation. It will be also relevant to discuss the predicted local structure at the site of interaction with proteins displaying enriched sequence motifs: the proteins discussed in the text bind to single stranded RNA: is the local structure at these sites predicted to be open within the secondary structure of these elements? By the way, Sex-Lethal is a Drosophila protein without a homolog in human cells, although a number of poly-U binding proteins (including ELAV-like, HuD and related factors) share similar sequence motifs. Therefore the presence of a SXL motif should be explained in these terms in the text and in Figure 5B.

The revised manuscript discusses the validity of the RNA secondary structure prediction in several ways.

First, we now explain in the text how these structures have been identified based on their similarity (conservation) in sequence and structure across vertebrate genomes (subsection “Ribosomes and post-transcriptional regulators bind to or around conserved RNA secondary structures”).

Second, we mapped all 5,504 predicted 3’ UTR RNA secondary structures to a publicly available dataset generated using Parallel Analysis of RNA Structure (PARS), which includes experimentally measured sensitivities to double stranded and single stranded ribonucleases to assess RNA structure. The predicted structures in our set do show overlap with the predictions from the PARS method suggesting an enrichment for real RNA secondary structures (Figure 7—figure supplement 1). The overlap is even more substantial when considering only the structures that are bound by proteins in our dataset (N=1,808). We discuss these validation tests in the main text (see the aforementioned subsection) and supporting material (Figure 7—figure supplement 1).

Third, while structure predictions do not capture the complexity that occurs in vivo, we argue that agreement between different structure prediction tools, in particular those that include experimental data, increase the confidence in correct prediction of the overall structure. Therefore, we examined our primary example, *HSP90B1*, more closely by incorporating SHAPE reactivities into the minimum free energy folding algorithm. The SHAPE-based and predicted structure was consistent. We discuss these comparisons with SHAPE in the main text (subsection “High-resolution data generate hypotheses on new regulators of the stress response and disease-relevant pathways”) and include the adjusted predicted structure (Figure 7C).

Lastly, we edited the text to emphasize that the structures are predicted and individual structures would need further validation by in vivo studies to truly confirm the RNA structure.

All these tests are now discussed in the text (subsection “High-resolution data generate hypotheses on new regulators of the stress response and disease-relevant pathways” and throughout) and supporting material (Figure 7—figure supplement 1).

We also now clarify that there is no known SXL homolog in human (subsection “Ribosomes and post-transcriptional regulators bind to or around conserved RNA secondary structures”, Figure 7). We also now discuss in the text that *ELAVL1* is an RNA-binding protein that would share a similar motif (AU-rich). We edited the discussion on Pumilio binding to address how single/double strandedness may impact the ability of proteins to bind (subsection “High-resolution data generate hypotheses on new regulators of the stress response and disease-relevant pathways”).